# rec-YnH enables simultaneous many-by-many detection of direct protein–protein and protein–RNA interactions

Jae-Seong Yang[1], Mireia Garriga-Canut[1], Nele Link[1], Carlo Carolis[1], Katrina Broadbent[1], Violeta Beltran-Sastre[1], Luis Serrano [1,2,3] & Sebastian P. Maurer [1,2]

Knowing which proteins and RNAs directly interact is essential for understanding cellular mechanisms. Unfortunately, discovering such interactions is costly and often unreliable. To overcome these limitations, we developed rec-YnH, a new yeast two and three-hybrid-based screening pipeline capable of detecting interactions within protein libraries or between protein libraries and RNA fragment pools. rec-YnH combines batch cloning and transformation with intracellular homologous recombination to generate bait–prey fusion libraries. By developing interaction selection in liquid–gels and using an ORF sequence-based readout of interactions via next-generation sequencing, we eliminate laborious plating and barcoding steps required by existing methods. We use rec-Y2H to simultaneously map interactions of protein domains and reveal novel putative interactors of PAR proteins. We further employ rec-Y2H to predict the architecture of published coprecipitated complexes. Finally, we use rec-Y3H to map interactions between multiple RNA-binding proteins and RNAs—the first time interactions between protein and RNA pools are simultaneously detected.

[1] Centre for Genomic Regulation (CRG), The Barcelona Institute of Science and Technology (BIST), Doctor Aiguader 88, 08003 Barcelona, Spain. [2] Universitat Pompeu Fabra (UPF), 08002 Barcelona, Spain. [3] Institució Catalana de Recerca i Estudis Avançats (ICREA), Pg. Lluis Companys 23, 08010 Barcelona, Spain. These authors contributed equally: Jae-Seong Yang, Mireia Garriga-Canut. Correspondence and requests for materials should be addressed to S.P.M. (email: sebastian.maurer@crg.eu) or J.-S.Y. (email: jae-seong.yang@crg.es)

Discovering which proteins, protein domains, and RNAs directly interact is an essential prerequisite for understanding the principles by which cellular machines operate. Assays capable of detecting such interactions are in high demand as they generate both system-level overviews of cellular coordination and hypotheses that can be tested in subsequent mechanistic studies[1]. Yeast two-hybrid and three-hybrid (Y2H, Y3H) screens[2,3] enable the detection of putative, direct protein–protein and protein–RNA interactions under physiological conditions, respectively. In the past, the Y2H assay has been significantly enhanced. Y2H-Seq[4] coupled Y2H screening to a next-generation sequencing (NGS) readout to reduce workload and costs while increasing sensitivity. To this end, selected baits and thousands of preys are grown in an array format, individually mated and pooled. Prey pools are narrowed down by NGS and finally interactions are detected by massive spotting assays on selective media. BFG-Y2H[5] first requires the creation of barcoded, arrayed libraries of baits, and preys. After pooled mating of barcoded prey and bait libraries and selection, intracellular Cre-recombination is induced to generate fused barcode tags that can be read by paired-end NGS in order to identify interacting pairs. Recently, CrY2H-Seq[6] used individual Gateway cloning of bait and prey libraries and subsequent individual transformations in yeast strains to generate screening libraries. After pooled-mating and intracellular fusion of interacting pairs by Y2H-reporter-induced Cre-recombination on plates, interactions are read out by NGS with a low number of reads containing information about interacting protein pairs. Compared to BFG-Y2H, CrY2H-Seq uses protein coding sequences to identify interacting protein pairs, avoiding the need for laborious library barcoding. While the discussed techniques offer significant advances, costly arraying[4–6], barcoding steps[5], or one-by-one subcloning and transformation of screening library constituents[4,6] are still frequently required. Furthermore, all existing techniques require library plating steps during interaction screening. Improvements of Y3H screens, however, have mostly focussed on optimising hybrid-RNA binding and presentation[7,8]. To date, no method allows the many-by-many detection of direct interactions between full-length RNA-binding proteins (RBPs) and RNA fragments.

By advancing on and integrating past developments of the Y2H and Y3H technologies, we developed a novel assay termed rec-YnH (recombination-YnH) that is capable of screening protein libraries against protein libraries or RNA fragment libraries. We focussed on keeping the workflow as simple as possible to make it both affordable and feasible. We combined batch transfer of DNA into screening vectors with batch library transformation, highly efficient DNA assembly by homologous recombination in yeast, liquid–gel culturing, and NGS-based readout of interactions using ORF sequences as identifiers to avoid any arraying, plating, or barcoding steps. As the endogenous yeast recombination machinery is used to fuse bait and prey, readily available yeast strains can be used for the screen, eliminating the need to create or obtain new strains expressing Cre-recombinase[5,6]. Furthermore, rec-YnH produces 40–50% reads containing interaction information, thereby using sequencing capacity comparably efficient. We demonstrate that the same assay pipeline can be used for rec-Y2H and rec-Y3H screening, creating for the first time an assay applicable to both, many-by-many protein–protein and protein–RNA interaction detection. We aimed to develop a tool especially useful for research labs focussing on mechanistic studies. Hence, we focused on molecular level questions involving 10s–100s of factors while aiming for maximal specificity to avoid false positives.

Using a well-characterised test matrix of proteins, we first benchmark rec-Y2H performance. Then, we demonstrate rec-YnH by focussing on problems typically faced by biomedical research laboratories: (i) understanding the interactions between hundreds of proteins in a given pathway, (ii) mapping the interactions of multiple protein domains, (iii) predicting the putative architecture of a coprecipitated complex, and (iv) detecting the RNA targets of a number of RBPs at once. rec-YnH enables us to validate interactions between PAR (partitioning defective) protein domains[9] and uncover novel putative interactions between PAR proteins, kinesin-cargo adaptors, and a dynein regulator. Furthermore, we discover new putative interactions between specific microtubule end-binding proteins (EBs) and plus-end-tracking proteins (+TIPs)[10], and show that quantitative measures of protein abundance obtained via mass spectrometry (MS)[11] depend on the number of nodes between baits and preys. Finally, we use rec-Y3H to precisely and reproducibly detect sparse true positive protein–RNA interactions among hundreds of possibilities. This is the first time that high-throughput screening is adapted and used for simultaneous multi-domain interaction mapping and multi-protein–RNA interaction mapping.

## Results

**rec-YnH assay design and workflow.** We developed three compatible screening vectors (pAWH, pBWH, and pMS22H) (Fig. 1 and Supplementary Table 1) based on widely available Y2H[2] and Y3H systems[7]. pAWH (prey vector) is used in combination with either pBWH (protein bait vector) or pMS22H (RNA bait vector) for rec-Y2H or rec-Y3H screens, respectively. While pBWH and pMS22H carry the TRP1 auxotrophic marker, pAWH carries the 2 μ yeast origin of replication (ORI). All three vectors also carry homology regions, which become exposed after digestion with homing enzymes. Both pAWH and pBWH are Gateway compatible and recombine efficiently in yeast (Supplementary Fig. 1). Batch Gateway LR reactions (Fig. 1 and Supplementary Fig. 2) generate hybrids of the ORFs fused to either the Gal4 activation domain (AD, pAWH) or the Gal4 DNA-binding domain (BD, pBWH). For rec-Y3H screening, RNA fragment subcloning by Gibson assembly (Supplementary Fig. 3) into pMS22H creates hybrids of RNA fragments fused to MS2-binding stem-loop sequences.

After digestion at two homing enzyme sites, linear pAWH and pBWH libraries (for rec-Y2H screens) or pAWH and pMS22H libraries (for rec-Y3H screens, Supplementary Fig. 4) are batch transformed into the yeast strains Y2HGold (Clontech) or YBZ-1[8], respectively (Supplementary Table 2). Then, transformed yeast cells are suspended in –Trp liquid–gel medium. Homologous recombination between one TRP1-containing fragment and one 2 μ yeast ORI-carrying fragment generates bait–prey fusion plasmids, enabling the host cell to propagate in –Trp recombination–selection (RS) medium. The remaining linear backbone fragments are unable to propagate in yeast because they lack both an auxotrophic marker and an ORI. As plating of clone libraries is costly, work intensive, and scalable only within limits, we adapted liquid–gel culturing for rec-YnH screening using low-melting agarose[12] (Supplementary Figs. 4, 5). As colonies are embedded in a gel matrix and remain isolated, overgrowth of fitter clones is prevented. At the same time, this procedure enables 3D culturing, thereby permitting the growth of ~1 million clones per 100 ml of medium. While performing as well as screening on-plate (Supplementary Fig. 6a, b), liquid–gel culturing is significantly less laborious and costly than harvesting colonies from plates (Supplementary Tables 3 and 4), permitting better scalability. Each transformed screen library is split into two cultures. While one culture grown in –Trp medium selects for library recombination (RS) to keep track of sampling complexity and to allow correction for clone frequency later on, the other

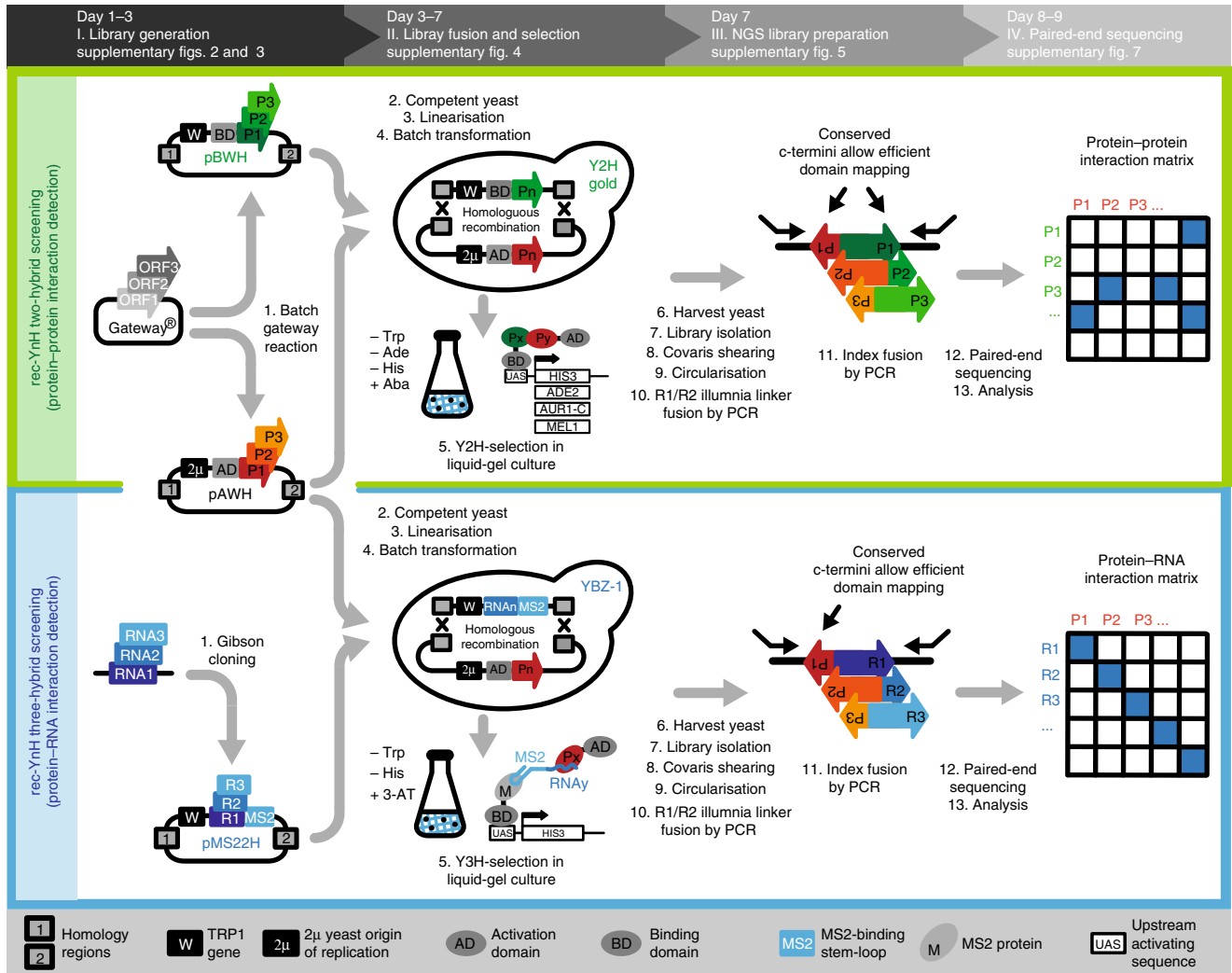

**Fig. 1** rec-YnH workflow. (I) ORF-coding or RNA-coding fragments are transferred into rec-Y2H or rec-Y3H screening vectors by batch Gateway or Gibson reactions (see Supplementary Figs. 2, 3). This step is done only once; the resulting libraries are used for several full-screen replicates. (II) pBWH and pAWH libraries (bait and prey rec-Y2H screening libraries) or pMS22H and pAWH libraries (RNA and prey rec-Y3H screening libraries) are linearised by homing enzyme digests and co-transformed into yeast in batch. The transformed yeast pool is split in a recombination–selection (RS) medium (–Trp, omitted in the figure for simplicity; see Supplementary Fig. 4 for details) and a recombination–interaction–selection (RIS) medium (–Trp and different selection markers available, see Supplementary Table 2). rec-YnH screening is done in liquid–gel cultures. Only correctly fused vectors contain a yeast origin of replication and a TRP1 marker, and are thus able to grow in RS or RIS media. (III) Cells are harvested by centrifugation, the fused plasmid libraries are isolated, and the DNA is fragmented by Covaris and re-circularised by intramolecular ligation. In a first PCR reaction, circular fragments containing the 3′ ends of both a bait and a prey are specifically amplified from the pool of fragments, adding R1/R2 Illumina adaptors. A second PCR step then adds a multiplexing index and P5/P7 Illumina attachment sequences, thereby creating a library of fused bait-coding and prey-coding sequences with conserved C-termini (see Supplementary Fig. 5 for details). (IV) Paired-end sequencing is used to readout library complexity and clone representation (RS condition) and bait–prey interactions (RIS condition). An analysis pipeline normalises the obtained reads, corrects for sequencing depth artefacts, and overrepresented clones (see Fig. 2 and Supplementary Fig. 7)

culture is used for recombination–interaction–selection (RIS) based on several reporters (Supplementary Table 2 and Supplementary Fig. 4). After culturing and harvesting by centrifugation, recombined plasmid DNA libraries are isolated in batch (Fig. 1 and Supplementary Fig. 5). By Covaris shearing to a length of 1200–1500 bps, intramolecular circularisation and PCR amplification, a pool of DNA fragments is generated containing each one bait and one prey-coding fragment. The chimeric fragment library is then analysed by paired-end sequencing (Supplementary Fig. 7). Differences between rec-Y2H and rec-Y3H are summarised in Supplementary Table 5.

**rec-YnH data analysis.** To benchmark rec-Y2H performance, we assembled a library of 76 human proteins (referred to as X76)

with well-characterised interactions[5] (Supplementary Data 1) and conducted five independent replicates (starting at the library transformation stage, Supplementary Data 2). Due to the high efficiency of homologous recombination of linear DNA[13] (Supplementary Fig. 1), and the selective amplification of fused fragments by PCR (Fig. 1 and Supplementary Fig. 5), rec-Y2H yields on average 40.7% (Fig. 2a) usable reads with mapped interaction pairs. This fraction of usable reads is considerably higher than those observed for existing techniques[6] (Supplementary Data 2). By cloning the X76 library only into the BD-containing pBWH vector and conducting an AD-empty screen (Fig. 2b), we were able to detect which proteins auto-activate the Y2H reporters. In contrast, by cloning the X76 library only into the AD-containing pAWH vector and screening against the empty pBWH vector

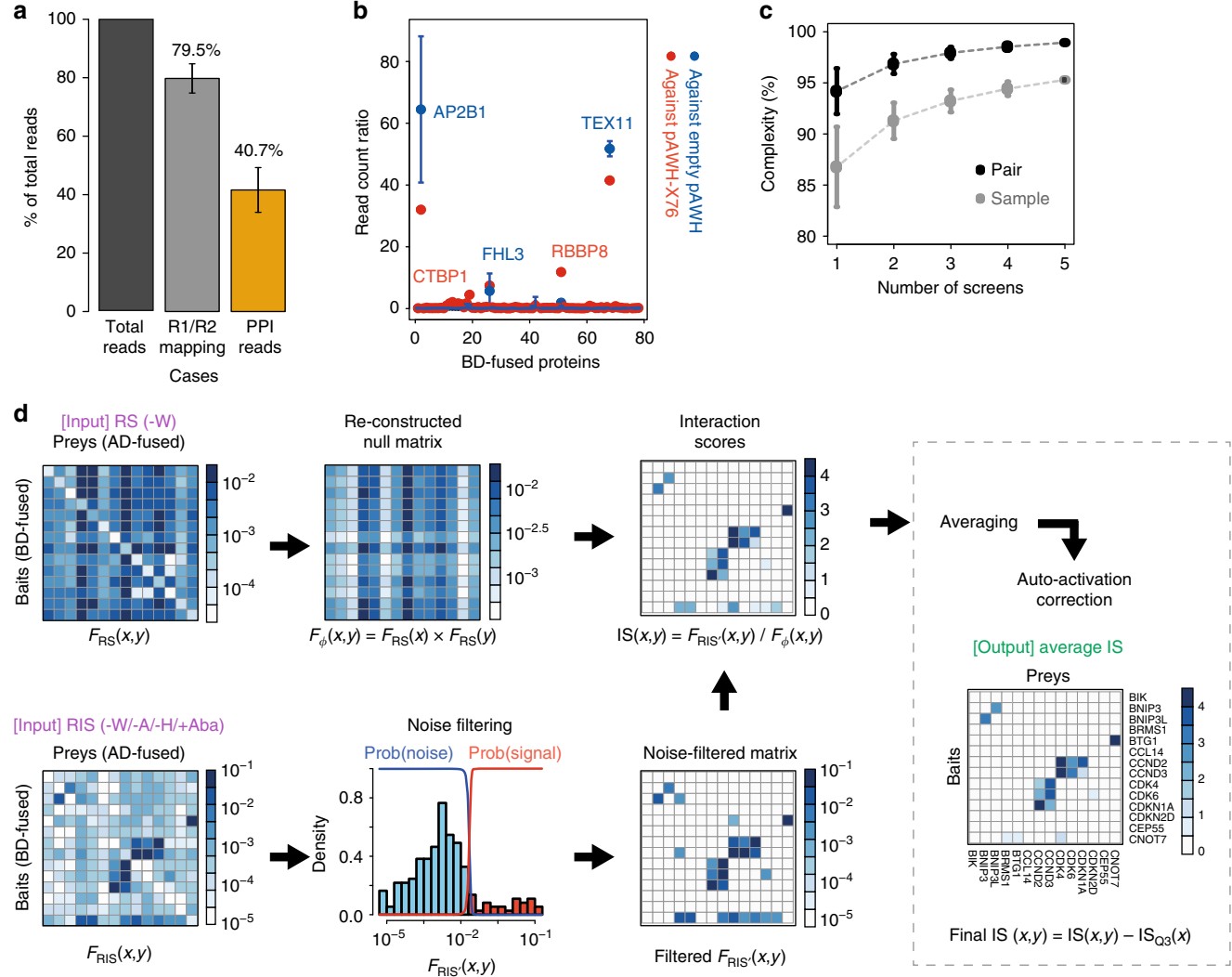

**Fig. 2** rec-YnH data analysis. **a** Estimation of rec-Y2H read mapping efficiency. On average, 79.5% of total reads are mapped with R1/R2 PCR primers (Supplementary Data 2), and 40.7% of reads are usable to map interaction pairs. **b** A priori detection of auto-activators. As a measure of the auto-activator signal, we calculated the ratio of bait protein-mapped read counts between RIS (recombination–interaction–selection) and RS (recombination–selection) conditions for an AD-empty (pBWH X76 pool/pAWH empty) and a full (pBWH X76 pool/pAWH X76 pool) screen. Blue circles indicate bait signals from the AD-empty screening, while red circles indicate bait signals in the full screen. **c** Sample and pair complexity as a function of the number of X71 library screens (X71 = the X76 library without auto-activators or toxic proteins; see Supplementary Fig. 8). Sample complexity is the fraction of pairs that form out of all possibilities under RS conditions, while pair complexity is the fraction of ORF combinations irrespective of the domain (i.e., BD or AD) to which they were fused. **d** Computational analysis pipeline of rec-YnH. The reconstructed null matrix is generated by normalising to total read counts and then to the frequency of bait–prey pairs under RS conditions ($F_{RS}(x, y)$). $F_{RS}(x)$ and $F_{RS}(y)$ indicate the row-wise and column-wise sum of the detection frequency $F_{RS}(x, y)$, respectively. Gaussian mixture fitting is used to filter out noise from the protein interaction signal of the RIS matrix ($F_{RIS}(x, y)$). The noise-filtered signal ($F_{RIS'}(x, y)$) is further normalised by the null matrix ($F_\phi(x, y)$) to generate the interaction score (IS) matrix. This corrects for clone overrepresentation. To remove stochastic false positives and increase sensitivity, several experiments are averaged, and a final averaged IS matrix is generated. Finally, basal auto-activation levels are removed by the subtraction of the upper quartile interaction score ($IS_{Q3}(x)$ = row-wise top 25 percentile IS) for each bait protein. Usually $IS_{Q3}(x)$ is 0 for non-auto-activator proteins. The same pipeline was applied to all rec-YnH screens done here. **a–c** Error bars represent standard deviations

(Supplementary Fig. 8a), we were able to detect which proteins are toxic. Also, we conducted one full screen (i.e., X76 cloned into both vectors) to compare the results obtained from the empty-vector pre-screens to those obtained with a full screen. As expected, the AD-empty screen detects auto-activation activity very well ($r = 0.95$, $P$-value $< 2.2 \times 10^{-16}$; Fisher Z-transformation, Fig. 2b), and the three auto-activators detected in pre-screening account for 54.8% of reads in the X76 full screen. We found two proteins (CTBP1 and RBBP8) with a high read count ratio in the full screen which were not detected in the AD-empty screen (Fig. 2b). They account for 7.9% of total reads and we

excluded these basal auto-activators as well for further screening. Removal of auto-activators was done by de novo assembling of a pENTR-ORF pool and batch Gateway cloning into screening vectors which requires 2–3 days. Hereafter, we call the X76 library without auto-activators fused to the BD, the X71 library. From the BD-empty screen, we also detected nine proteins with a toxic effect (Supplementary Fig. 8a–c). As rec-Y2H discriminates both AD and BD fused proteins, the same protein pairs can be detected twice. This has two advantages. It increases the level of confidence if a pair is found in both orientations and it allows to avoid auto-activation or toxicity-induced limitations. Using the RS controls

(i.e., –Trp conditions), we measured both sample complexity and pair complexity. While sample complexity measures how many of all possible bait–prey fusions were formed, pair complexity measures whether at least one of two possible fusions between two ORF-containing fragments was formed irrespective of the domain to which each ORF is cloned (i.e., Px-AD—Py-BD or vice versa). After five screens, sampling and pair complexity reach 95.3 and 98.9%, respectively (Fig. 2c), with no detectable differences between on-plate and liquid–gel culturing (Supplementary Fig. 6b, Supplementary Data 2).

A pipeline we developed to process the NGS data generated by rec-YnH screening (Fig. 2d) normalises matrices obtained from analysing sequencing data of RS and RIS conditions to correct for possible biases stemming from different input DNA amounts, fitness effects, library amplification, and sequencing depth limitations[5]. To remove experimental noise, we applied a Gaussian mixture-based noise filter[14] to the RIS matrix with the assumption that most of the protein pairs are not interacting with each other. Normalising the noise-filtered RIS matrix to the reconstructed null matrix yields a matrix of interaction scores (IS). Finally, matrices obtained from screen replicates are averaged to filter-out stochastic false positives and the row-wise upper quartile score is subtracted to compensate for basal auto-activation activity. Through the removal of false positives generated from experimental noise, our method improves correlation between experiments by 17.6% (Supplementary Fig. 8e, f, $P$-value $= 3.7 \times 10^{-12}$; two-tailed unpaired $t$-test) compared to using an existing processing method[5]. All scripts needed for rec-YnH data analysis are provided on GitHub (see "Data availability" for details).

**rec-Y2H assay performance**. Next, we benchmarked the performance of rec-Y2H according to published criteria[15]. Using direct protein interactions found in the BioGRID and HIPPIE protein–protein interaction (PPI) databases as true positive control values and defining all interactions not present as true negative values, we calculated an optimal, normalised read cut-off value of 1.2 based on the F1-Score and 2.8 based on the Matthews Correlation Coefficient (MCC) (Fig. 3a). Using the F1-Score cut-off value, we were able to detect 163 positive interactions, of which 38.6% are found in PPI databases (Fig. 3b and Supplementary Data 3). Retesting 207 randomly chosen positive and negative hetero-interaction pairs in individual spot tests (Fig. 3c, Supplementary Fig. 9) yielded an excellent correlation with rec-Y2H (sensitivity 95.5%, specificity 99.2%, and AUC value of 0.96, Fig. 3d and Supplementary Data 4d). This corresponds to a retest rate of 98.8% (40/40 novel interactions and 44/45 known interactions, Supplementary Data 4a). Four out of 122 interactions not detected by rec-Y2H were detected by spot tests and among them 3 have rec-Y2H scores above 0 but below F1-Score-based cut-off. Using an MCC-based cut-off value, in contrast, we observed that the sensitivity relative to spot tests was decreased to 73.9% (Supplementary Data 4d). We hypothesised that the MCC gives a higher cut-off value because we used PPI databases including non-Y2H-based evidence as reference set and hence decided to define cut-offs based on the F1-Score. When we used only our Y2H spot tests as a reference set, we obtained a cut-off value of 1.3 based on F1-Score and MCC (Supplementary Fig. 10). Upon comparing averages and individual experiments to spot test results, we found that averaging multiple experiments increases sensitivity by 11.2% while only causing minor changes in specificity (Fig. 3e and Supplementary Fig. 11). By analysing the fraction of all detectable interactions that we found in each X71-screen replicate (total nine interaction selection conditions), we determined a mean sampling sensitivity of 86.0% ± 7.0% (Fig. 3f).

Therefore, while simplifying the assay workflow, rec-Y2H is able to perform on par or even better than other currently available methods (Supplementary Data 4b–d)[5,6,15].

We also tested 30 positive and negative homo-interaction pairs. A high fraction of homodimers (8/9) which were not detected by rec-Y2H were detected in spot tests. We noticed that rec-Y2H detects homodimers less well due less efficient PCR amplification of homodimers that presumably form long hairpin structures (coming from palindromic sequences of the same 3′ end), hampering library preparation and NGS[6] (Supplementary Fig. 12). Hence, we excluded homodimers in our performance analysis.

**Mapping of interactions between PAR protein domains**. Next, we wanted to apply rec-Y2H to confirm and extend our knowledge of interactions between PAR proteins, their domains, motor proteins, and microtubule cytoskeleton-organising proteins. PAR proteins are conserved polarity regulators[9] whose localisation, for example, control axon determination during neuronal differentiation[16]. However, little is known about how PAR proteins are localised in neurons or how they affect microtubule regulation[16]. Furthermore, we were interested in understanding if and how different EB homologues recruit different +TIP networks[10]. Expression and localisation of EB homologues differs during neuronal differentiation[17,18], and it is speculated that they carry out distinct functions by recruiting different regulatory +TIP networks to microtubule ends[11,19]. Nonetheless, the extent to which direct EB interactomes differ is not understood. To shed light on these questions, we assembled an ORF library that, after removing auto-activators, contained 163 mouse and human proteins (Supplementary Data 5, hereafter termed X163). The X163 library contains full-length clones and domains of PAR proteins[9], a number of microtubule-based motor proteins and motor-cargo adaptors. Finally, we added 98 proteins found to interact with three homologues of microtubule + TIPs (EB1, 2, and 3) in an affinity capture MS study[11]. This latter group contains several microtubule-associated proteins (MAPs) that control microtubule organisation.

Reliable domain mapping with NGS-based readout requires that at least one terminus of a peptide-coding fragment remains detectable after the screening procedure. Since rec-Y2H conserves the 3′ end of the peptide-coding insert in a high fraction of sequencing reads (Fig. 2a), we anticipated that our assay would be ideal for simultaneously mapping interactions between multiple protein domains—a task which is traditionally slow and low-throughput. Figure 4a shows the fraction of the X163 interaction-matrix containing full-length PAR proteins and their domains, and Fig. 4b the derived interaction scheme of full-length PAR proteins. Compared with interactions collected from the literature, rec-Y2H did not detect three interactions between full-length PAR proteins: (1) MARK2-PRKCI, (2) PARD3-PARD6G, and (3) PARD3-PRKCI. Interaction (1) is a phosphorylation reaction and thus binding might be too transient for detection, (2) was potentially not detected due to auto-inhibition of full-length PAR3[20] or due to the lack of a PAR3-fragment containing the full PAR6 interaction domain[21], (3) on the other hand, was detected but fell below the F1-Score-derived cut-off value (Fig. 4b and Supplementary Fig. 13). Figure 4c shows both known and rec-Y2H-detected interactions between PAR proteins and their specific domains, all of which are confirmed in the literature[9,22]. For most cases interactions where detected for both the full-length proteins and their domains (Fig. 4a), while the known PAR1–PAR3 interaction[23] was detected only at the domain level (Fig. 4a, c). This could stem from intramolecular auto-inhibition mechanisms reported for PAR3[20], and highlights the complementary value of screening for domain–domain

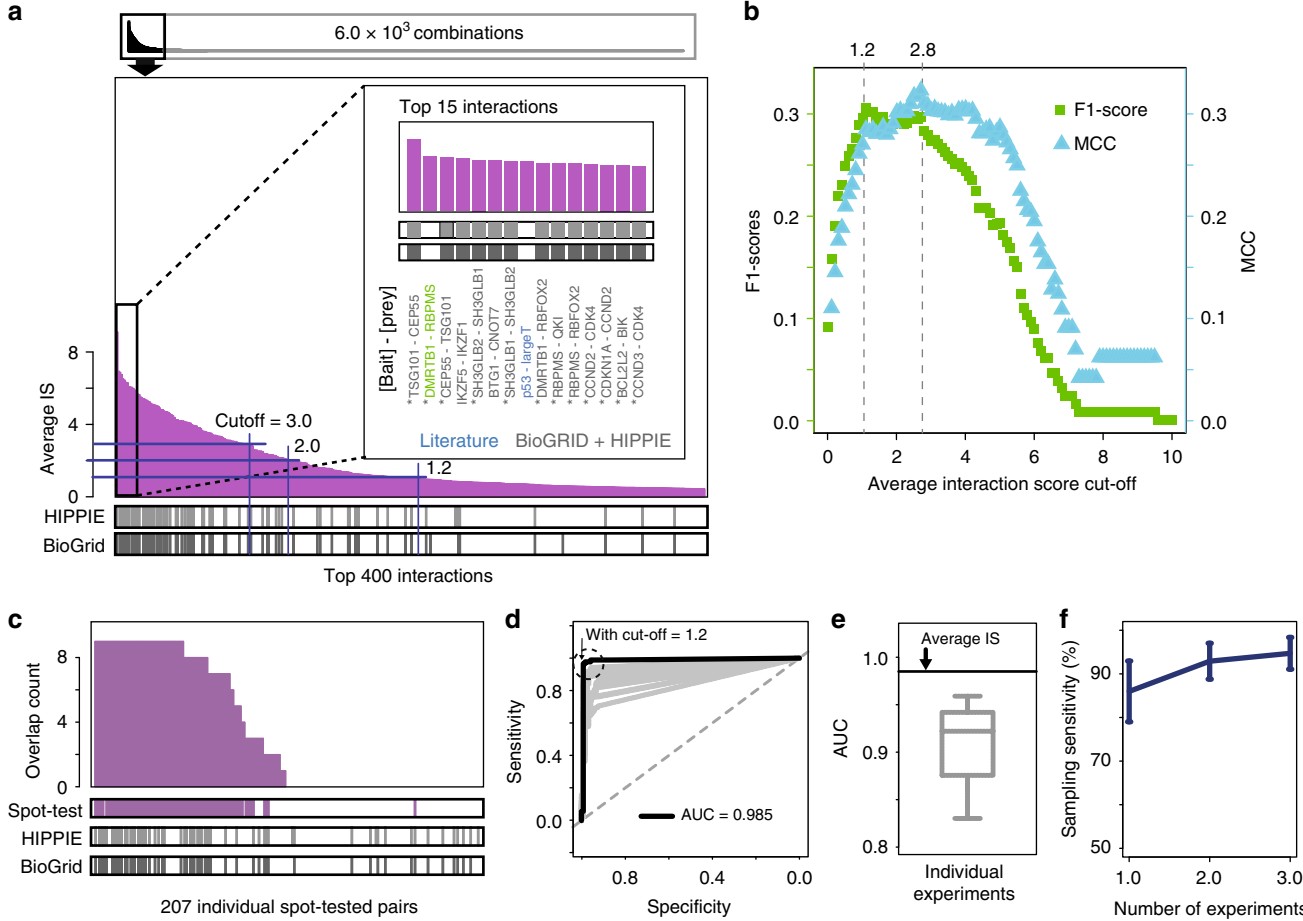

**Fig. 3** rec-Y2H assay performance. **a** The top 400 protein pairs from the X71 screen having the highest average interaction scores (average IS). Bars at the bottom indicate if interactions were annotated in protein–protein interaction (PPI) databases. Top 15 pairs are further highlighted in the inset. Among them, 13 were found in PPI databases (grey). Asterisk indicates interactions detected above the cut-off (average IS = 1.2) in both orientations (Px-AD—Py-BD or vice versa). Blue indicates interactions not found in PPI databases, but described in the literature. Green indicates an interaction previously not reported. **b** Determination of the optimal interaction score cut-off based on the harmonic average of precision and sensitivity (F1-Score, green) and Matthews's Correlation Coefficient (MCC, cyan). The performances were evaluated based on combined BioGRID and HIPPIE PPI databases. We stringently defined all pairs not found in both databases as none-interacting pairs. **c** Comparison of interaction pair detection counts across experiments, individual spot test results (see Supplementary Fig. 9) and PPI databases giving an overall pairwise retest rate of 98.8%. Two hundred seven pairs were tested by spot test without homodimers. **d** Overall performance (AUC) of rec-Y2H based on the results of 207 individual spot tests. The black and grey lines indicate the averaged and individual experiments, respectively. The grey-dotted diagonal line shows random performance. **e** Distribution of the AUC for individual experiments. The box of the box plot illustrates the median, upper, and lower quartiles for AUCs. The black line and grey box-plot represent averaged and individual experiment performance, respectively. **f** Mean sampling sensitivity of single experiment 86.0% ± 7.0% (average of nine individual experiments). The sampling sensitivity is already saturated after averaging just two experiments. Error bars represent standard deviation

interactions in addition to full-length proteins. PAR protein distribution in large cells such as neurons relies on active transport[24], and PAR proteins have also been shown to bind to and effect microtubule dynamics[20]. For both cases, only a few examples are known. The X163 library (Supplementary Data 5) was designed to deepen our understanding of which PAR-MAP/Motor interactions exist. We found 234 interactions (Supplementary Data 6) above the F1-Score-defined cut-off (Supplementary Fig. 13), with the highest ranking shown in Fig. 4d. Interestingly, all PAR proteins known to localise to axons[16] bind to CASK and KAP3 (=KIFAP3) (Fig. 4e), which are known cargo-loading adaptors for the axonal kinesins KIF1 and KIF3, respectively[25,26]. KIF3A was previously shown to transport Par3, or with a further adaptor, the aPKC–PAR6 complex[24,27] into axons. Aside from its function as a kinesin-cargo adaptor, CASK is known as a scaffold protein involved in the coordination of presynapse formation[28]. Hence, newly detected PAR–CASK interactions might indicate an alternative axonal transport

pathway for PAR proteins, or an involvement of axonally localised PAR proteins in the regulation of synaptogenesis. The newly detected PAR6–NDEL1 interaction is equally interesting; NDEL1 controls cytoplasmic dynein activity and polarised cargo transport in neurons[29]. By modulating NDEL1 interactions or function, PAR6 could contribute to the regulation of polarised cargo trafficking. To test how reliably rec-Y2H is predicting novel interactions, we re-tested 35 high-interest pairs containing novel PAR protein interactions. We chose the NanoBRET assay[30] for this, as it differs in many important aspects from rec-Y2H. NanoBRET is a bioluminescence resonance energy transfer (BRET)-based interaction assay that can detect interactions between reporter-fused proteins in the cytoplasm of mammalian cells. Limitations of NanoBRET are that reporters must be in close proximity (<10 nm) for signal induction and proteins need to localise to the cytoplasm. Still, with NanoBRET major possible causes for false positives in Y2H-based screens (BD-fused proteins recruiting RNA polymerase or AD-fused proteins

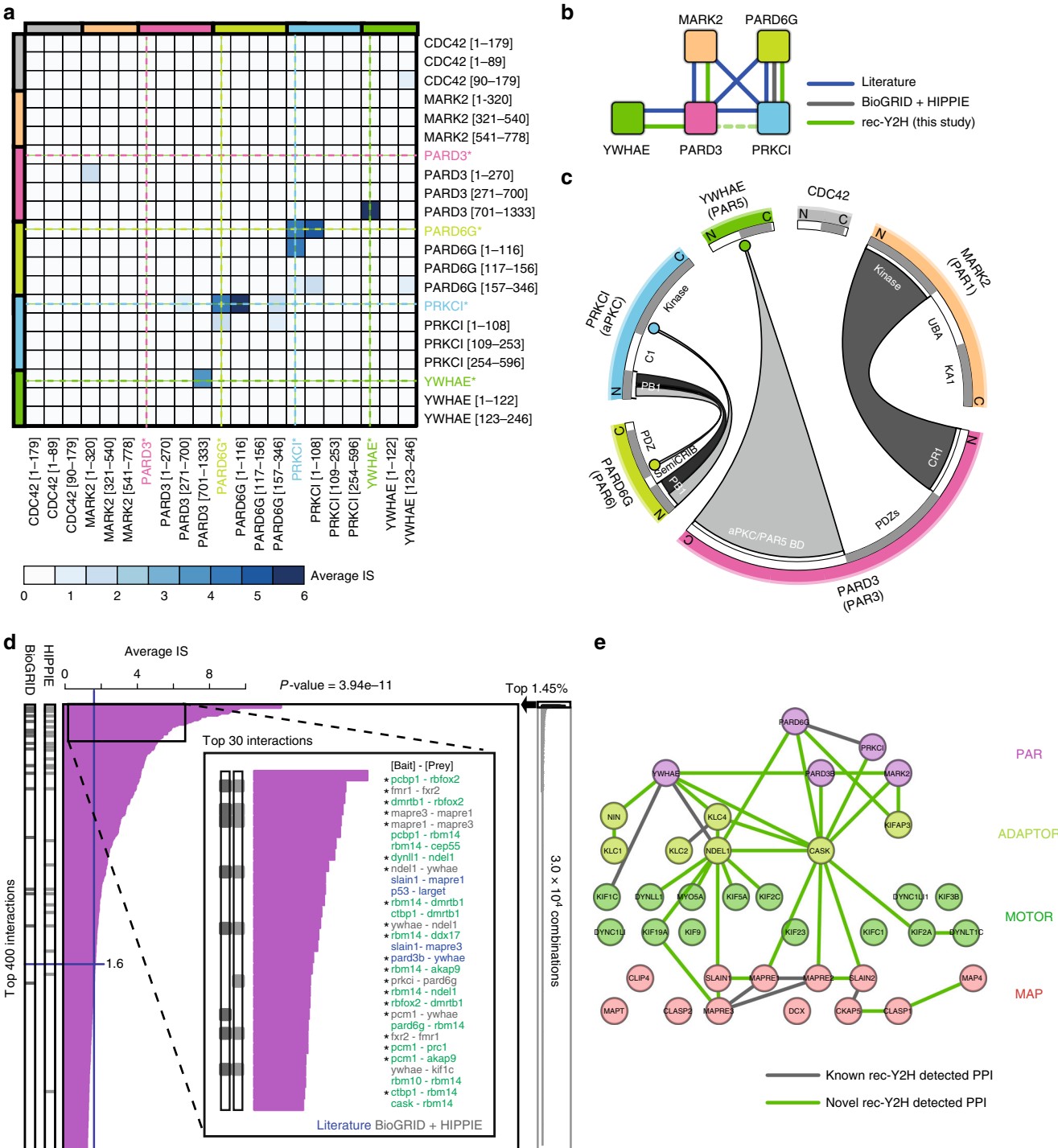

**Fig. 4** Simultaneous mapping of interactions between PAR protein domains. **a** Matrix showing the average IS between full-length PAR proteins and their domains. **b** Interaction scheme for full-length PAR proteins based on interactions from the literature, PPI databases, and data from our rec-Y2H screen. Dotted line between PARD3 and PRKI indicates that the average IS (0.95) for this interaction was below our cut-off of 1.6 (based on F1-Score; see Supplementary Fig. 11). **c** Circular representation of the detected interactions involving PAR protein domains. Coloured circles situated in the middle of a protein represent interactions between a PAR protein domain and a full-length protein. Interactions detected between two full-length proteins (shown in **a**) have been omitted for clarity. **d** Top 400 protein pairs with the highest average IS compared to interactions annotated in PPI databases. *P*-value indicates how many known PPI are enriched in top 400 by hypergeometric test. The top 30 pairs are further highlighted in the inset box. Among them, nine are found in PPI databases (grey), and four are positive controls or literature-curated interactions (blue). Among the newly found 17 pairs (green), 11 pairs were detected in both orientations (Px-AD—Py-BD or vice versa, marked with asterisk), and are thus of high confidence. The rightmost box indicates the possible $3.0 \times 10^4$ interaction pair combinations. **e** Interaction hierarchy of the PAR-MAP network. All the interactions depicted here were detected by rec-Y2H. Interaction pairs also annotated in the PPI databases are shown as grey lines

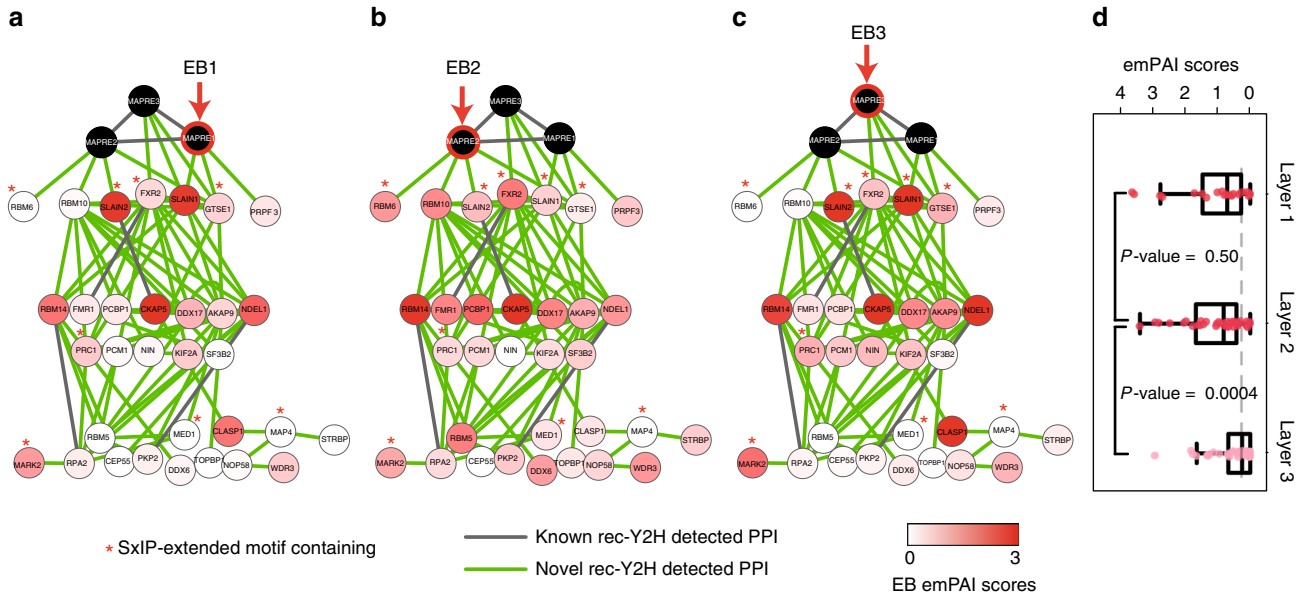

**Fig. 5** rec-Y2H refines the putative architecture of coprecipitated complexes. **a–c** Direct interaction networks of the microtubule end-binding proteins (EBs). The first layer represents the direct EB-binding partners found in this screen. The second and third layers represent indirect binding partners, separated from the EBs by one or two nodes. All interactions depicted here were detected by rec-Y2H. Interaction pairs also annotated in PPI databases are shown as grey lines. Proteins having known EB interaction motifs (SxIP) are marked with a red asterisk. We used an extended SxIP-9AA motif definition with the following rules: (1) $X_1 - X_2 - [ST] - X_3 - [IL] - P - X_4 - X_5 - X_6$; (2) $X_1 - X_4$, at least one basic amino acid (R/H/K); and (3) $X_1 - X_6$, no acidic amino acid (D/E)[11]. Nodes are coloured according to the protein abundance in the pull-down complex (emPAI score)[11] with **a** EB1 as bait, **b** EB2 as bait, and **c** EB3 as bait. In all cases, we could not find the alternative LxxPTPh EB interaction motif[32]. **d** emPAI scores for each layer. The box of the box plot illustrates the median, upper, and lower quartiles for emPAI scores. The data from all three EB protein pull-downs were combined in the box plot representation and each data point is represented as a dot. The grey-dotted line indicates the median emPAI score of Layer 3 proteins. P-values were calculated by Wilcoxon rank-sum test

binding to DNA) can be ruled out. Furthermore, screening in a different background reduces the chance of confirming false positives caused by "bridging proteins". Overall, we could validate 60.0% of interactions (21/35) (Supplementary Fig. 14a), with a higher fraction of validated interactions among pairs with high average IS (Supplementary Fig. 14b). Furthermore, 11 out of 13 interactions found in both orientations (AD-BD or BD-AD fused) were validated, confirming these screen read outs as useful confidence indicators.

**rec-Y2H refines the architecture of coprecipitated complexes.** In biomedical research, affinity-capture-MS analysis is often used to investigate protein interactions. This analysis, however, informs only about physical but not direct interactions. Having a complementary tool available, capable of predicting how proteins in an isolated complex directly interact, would be a significant advantage. With the aim of understanding better the hierarchy of interactions among proteins coprecipitated with the three EB homologues, we analysed the interaction networks between EBs and proteins found in published data[11]. We found that of the 98 EB-coprecipitated proteins we tested within the X163 set, only 7 directly bind to EBs (Fig. 5a–c and Supplementary Data 6) while we failed to detect some previously reported interactions such as EB1-KIF2C[31]. While five of these seven direct interactors possess known EB interaction motifs[11,32], only 16% (4/25) of the indirect interactors carry one (P-value = 0.01; Hypergeometric test). Nonetheless, as proteins not containing any known EB interaction motif were found to bind directly to EBs (RBM10, PRPF3), it is possible that unknown EB-binding motifs exist. Compared to EB2, for which three out of four direct connections are distinct interactors, EB1 and EB3 share two out of three direct connections. This is in line with published evidence, showing that EB1

and EB3 have similar deletion phenotypes and more readily hetero-dimerise compared to EB2[19]. The published emPAI scores, which are a measure of protein abundances in MS samples agree with the results from our rec-Y2H assay. For example, RBM6 and RBM10, which were found to directly interact only with EB2, are not found or had a low emPAI score when EB1 or EB3 were used as baits. These proteins had high emPAI scores when EB2 was used as bait. Of note, we were able to confirm the EB2–RBM6 interaction by NanoBRET, while neither EB1 nor EB3 were interacting with RBM6 (Supplementary Fig. 14a). This adds confidence in the precision of rec-YnH while predicting EB-specific interactions with certain RBPs. In general, the emPAI scores decrease significantly from the second to the third layer of interaction (Fig. 5d, P-value = 0.0004; Wilcoxon rank-sum test), suggesting that the interaction hierarchy predicted by our screen reflects the organisation of the pulled-down complex to some extent. Finally, we note that putative binary interactions found using rec-Y2H could account for 32.7% of EB-coprecipitated proteins we tested, a value significantly higher than the ones obtained using current PPI databases (12.2%, Supplementary Fig. 15).

**Multi-RNA–protein interaction mapping by rec-Y3H.** To further demonstrate the versatility of rec-YnH, we assembled a benchmarking library (RNAlib) consisting of 17 RNA targets and 19 proteins (Supplementary Data 7). This included 16 ribonucleoprotein complexes, selected based on previously published co-complex structures[33], the IRE/IRP pair as a positive control, and MBP and GFP as negative controls with no known RNA-binding activity. The selected proteins and protein fragments possess diverse RNA-binding domains (Fig. 6a). We screened the RNAlib four times, including two full and two technical replicates, and

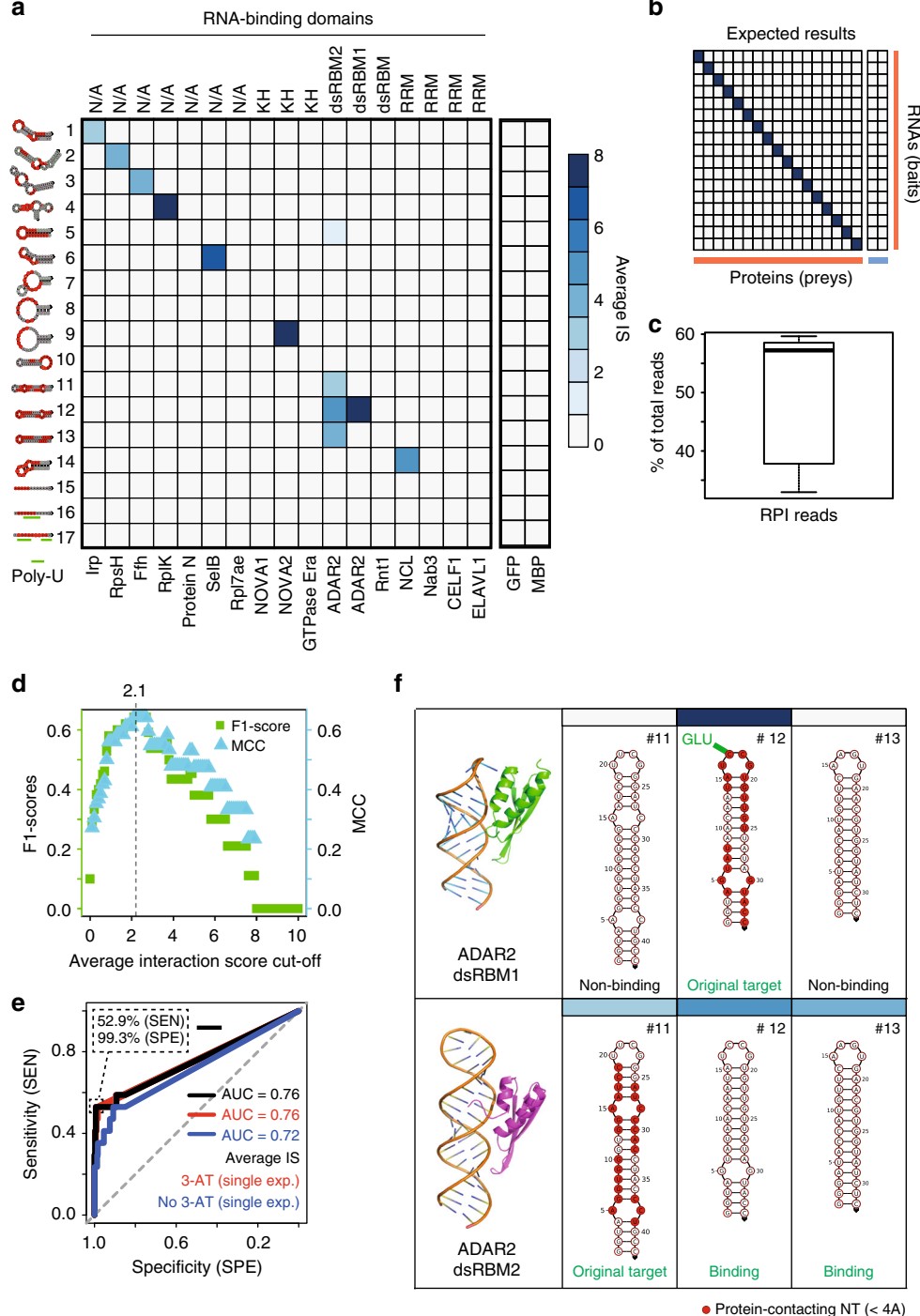

**Fig. 6** Simultaneous multi-RNA–protein interaction mapping by rec-Y3H. **a** Interaction matrix showing the average IS between 17 RNAs and 19 proteins. The RNA–protein pairs were selected from published RNA–protein complex structures[33]. GFP and MBP are negative controls, while IRP-IRE (RNA #1) is a positive control (see Supplementary Data 7 for full list). On the left, secondary structures of the screened RNAs are represented. Red-coloured dots indicate protein-contacting nucleotides (<4 Å). Green lines represent Poly U sequences ≥4 consecutive Us, which cannot be detected by RNA polymerase III-based Y3H screens. RNA-binding protein names are represented along the bottom. RNA-binding domain class is specified at the top when applicable. **b** Expected RNA–protein interaction map[33]. **c** Estimation of rec-Y3H read mapping efficiency. The box of the box plot illustrates the median, upper, and lower quartiles for read mapping efficiencies. On average, 50.5% of reads are usable for mapping interaction pairs. **d** Optimal interaction score cut-off based on F1-Score (green) and Matthews's Correlation Coefficient (MCC, cyan). **e** Overall performance (AUC) of rec-Y3H based on the expected 17 diagonal positive and 306 off-diagonal negative interaction pairs. The black, red, and blue lines indicate the averaged and individual (with and without 3AT) experiments, respectively. The grey-dotted diagonal line shows random performance. The inset shows assay sensitivity and precision at an optimal cut-off of 2.1, as defined based on the F1-Score in the preceding panel. **f** NMR structures of two dsRBMs that show distinct behaviour in rec-Y3H. dsRBM1 binds close to the apical loop, contacting specific unpaired bases present only in the loop of RNA #12, while dsRBM2 less specifically binds along the double-stranded stem

subsequently validated the results with one-on-one spot tests (Supplementary Fig. 16a, b). Addition of 3-AT significantly increased the signal-to-noise ratio of the final result (Supplementary Fig. 16c). After signal normalisation (as done for the rec-Y2H screen; Fig. 2d), we obtained an RNA–protein interactions matrix (Fig. 6a) and compared it to the expected outcome (Fig. 6b). With on average 50.5% reads usable for RNA–protein interaction mapping (Fig. 6c), rec-Y3H performs as well as rec-Y2H (Fig. 2a). Selecting a cut-off value based on the F1-Score (Fig. 6d) yielded 11 positive interactions with a specificity of 99.3% and a sensitivity of 52.9% (Fig. 6e). While for eight test proteins we unambiguously detected a specific RNA interaction, in one case (dsRBM2-domain of Adar2), multiple RNA interactions were detected. Intriguingly, all of the dsRBM2-bound RNAs exhibited base-paired stems similar to the original dsRBM2 target (Fig. 6f, lower row). At the same time, the dsRBM1 domain, which binds RNA close to the apical loop, was found to interact solely with its known target RNA. Close inspection of RNAs #11 and #13 (Fig. 6f, upper row) shows that a crucial glutamate–cytosine contact[34] is not possible at the apical loops of these RNAs. To validate these hypotheses further, we mutated RNA contacting amino acids of dsRBM1 and dsRBM2 domains. The dsRBM1-E16A mutation (Supplementary Fig. 17a) affects the glutamate contacting the already mentioned cytosine in the loop of RNA #12 (Fig. 6f, lower row). This mutation interestingly leads to a loss of specificity of dsRBM1 (Supplementary Fig. 17b–c), supporting the idea that specificity of this domain is controlled by both, loop and A-form helix contacts of dsRBM1. Mutating the minor-groove contacting amino acids of dsRBM2 (S29A, H30A) leads to an almost complete loss of binding to all three RNAs (RNA #11, #12, and #13) as expected (Supplementary Fig. 17b–c). Of the seven RNAs not found to interact with any of the proteins, two of them carry poly-U stretches that lead to transcription abortion of RNA polymerase III and cannot be detected by Y3H-based assays[3]. The lowest affinity detected was 1.1 μM; two more of the non-detected cases (RNAs #13, #15) have affinities in the >10 μM range[33] (Supplementary Data 7), which rec-Y3H might not be able to detect[3].

## Discussion

rec-Y2H is the least-laborious, low-cost (Supplementary Table 6), high-precision direct PPI screening method available to date. Furthermore, rec-Y3H enables screening of protein libraries against RNA fragment libraries, thereby providing the first many-by-many protein–RNA interaction screen. At the same time, it solely requires our newly developed vectors, widely available yeast strains, the accompanying published Supplementary Methods, and the analysis script to implement rec-YnH screening in any standard laboratory. We highlight the broad applicability of rec-YnH by addressing four common problems. As the aim of our study was to demonstrate the reliability of rec-YnH for molecular-level questions, our test libraries ranged from $10^2$ to $10^4$ combinations. However, due to the possibility of detecting auto-activators a priori, and our comparably high fraction of usable NGS reads, sequencing capacity is not likely to become a limiting factor for scaling-up (Supplementary Fig. 18). Additionally, as screening in liquid–gel cultures theoretically offers great scalability, rec-YnH potentially allows very large-scale screens but this remains to be tested in the future.

Despite all improvements, rec-YnH is still bound to the limitations imposed by Y2H and Y3H techniques. For instance, some interactions are not detectable due to a lack of certain PTMs in yeast, low protein expression, or toxicity of expressed proteins. As shown here, one way to circumvent toxicity issues is to split proteins into domains. Another possibility, which occurs by default during rec-Y2H screening, is to swap baits and preys. Another limitation of Y3H-based screens is that motifs with more than four consecutive uridine residues cannot be detected[3]. Nonetheless, we anticipate that the sensitivity of rec-Y3H screening can be further improved by using advanced designs of RNA target-presenting structures or alternative RNA polymerase promoters to drive expression of hybrid RNAs[3,7].

The procedural advances applied during rec-YnH screening (i.e., the liquid–gel culturing and noise filtering of raw data) can be combined with existing techniques[5,6] to possibly improve their performance further. The new putative direct interactions we detected between PAR proteins, motor adaptors and regulators, or EB proteins and +TIPs raise interesting questions such as whether PAR proteins regulate the activity of and/or are transported by different motor proteins in mammalian cells. The EB-specific, putative interactions we detected and partially validated between EB2 and the RBPs RBM6 and RBM10 support the notion that interactomes of EB-homologues differ. The putative direct interactions between EBs and RBPs we detected further raise the question of whether the microtubule cytoskeleton not only functions as a highway to transport RNPs—eventually EBs at dynamically growing microtubule plus-ends undergo interactions with RBPs, thereby connecting microtubule dynamics with RNP deposition or remodelling.

We envision that rec-Y2H will not only be used to understand the wiring of protein networks, but also to map interaction domains in a high-throughput manner. Using rec-Y2H as a complementary assay to MS analysis of coprecipitated complexes will significantly increase our level of knowledge by predicting which proteins in an isolated complex directly interact. The wide availability of ORFeome collections makes such a complementary use of rec-Y2H feasible. By combining multi-domain mapping with rec-Y3H screening, the RNA targets of RBP domains can be analysed in a high-throughput fashion. The same approach can be used for protein–RNA interaction perturbation mapping[35]. At the same time, the flexibility of the Y3H method in being able to screen RNA targets between ~20 and 200 nucleotides in length opens up the possibility to investigate sequence-context effects of RBP–RNA interactions on a systematic level. Lastly, the combination of rec-Y2H and rec-Y3H screening provides a tool to predict specific RNP structures from binary protein–protein and protein–RNA interaction data. This type of information promises to provide valuable insights into the regulation of mRNA localisation and expression.

## Methods

**Construction of three pDEST vectors for rec-Y2H**. See Supplementary Table 1 for detailed information of all vectors. pAWH (plasmid with Gal4 activation domain, gateway cassette and homology regions) was built in two steps. First, the attR1-ccdB-attR2 cassette (from pACT4-DM, courtesy of Erich E. Wanker's lab) was cloned into the BamHI site of pGADT7 (Clontech), to give pAW. Second, to introduce the homology regions 1 and 2, pAW was used as a template to generate two PCR fragments with linkers containing the 60-bp homology regions 1 and 2. We used 60-bp homologous sequences that have weak secondary structures and no homologous sequences to yeast genome to obtain high yield of recombination transformants[13]. The two fragments were then fused together in a Gibson assembly reaction to pAWH. On the other hand, two different pBWH (plasmid with Gal4 binding domain, gateway cassette and homology regions) vectors were constructed. (1) pBKWH, which contains a kanamycin resistance gene and is compatible with pENTR223. (2) pBSWHhc, which has a spectinomycin resistance gene and is compatible with pENTR221. First, a BglII fragment containing the attR1-ccdB-attR2 cassette (from pACT4-DM, courtesy of Erich E. Wanker's lab) was cloned into the BamHI site of pGBKT7 (Clontech), to give pBW. Second, to introduce the homology regions 1 and 2, pBW was used as PCR template and combined with either a kanamycin or a spectinomycin resistance cassette via Gibson assembly. This generated pBKWH and pBSWHhc, respectively.

**X76 and X163 library assembly by Gateway cloning**. See Supplementary Fig. 2 for details. To assemble the X76 library, pENTR223-ORF clones (see Supplementary Data 1) were picked from the Human Entry ORFeome v8.1 collection

(transOMIC) and plasmid DNA prepared using the QIAprep Spin Miniprep Kit (Qiagen). Each plasmid was diluted to 10 nM, and equal amounts of each clone were mixed to make an equimolar pENTR223-X76 pool. To assemble the X163 library (see Supplementary Data 5), 97 clones were picked from the Human Entry ORFeome v8.1 collection (transOMIC) to make a pENTR223-X163_a pool. The remaining 66 clones were not present in the Human Entry ORFeome v8.1 collection, and were generated by PCR from a mouse brain cDNA library, with primers specific to the gene of interest and attB linkers, and cloned into pDONR221 (Thermo Fisher Scientific) using BP clonase II enzyme mix (Invitrogen) according to the manufacturer's instructions. Equimolar amounts were mixed to generate the pENTR221-X163_b pool. Each pENTR pool was independently subcloned into pAWH and pBWH (pBKWH or pBSWHhc). For each pENTR pool, four Gateway LR reactions (5 μl each) were set up, containing 15 fmols of a pDEST vector (pAWH or pBWH), 15 fmols of pENTR-ORF pool, and 1 μl of LR clonase II enzyme mix (Thermo Fisher Scientific), following the manufacturer's indications. pDEST and LR clonase II enzyme mix were added every 8 h for two further cycles of incubation. Immediately afterwards, each LR reaction was transformed into NEB stable cells (New England Biolabs), and each transformation was spread on two LB-agar plates with the indicated antibiotic. Colonies were harvested by adding PBS to each plate, scraping, and then pooling together cells from eight plates. The OD of the bacteria pool was measured, and plasmid DNA was prepared with eight QIAprep Spin Miniprep Kit (Qiagen) columns using 15 OD units per column. DNA from the eight columns was mixed together and the concentration determined by $OD_{600}$. Each pDEST-ORF pool was diluted to 5 nM. pDEST X163_a and pDEST_163_b were mixed at an equimolar ratio to produce the final pDEST-X163 library. Each library is sufficient for performing 5–10 screen replicates, each with two selection media (RS and RIS).

**RNAlib protein library assembly by Gateway cloning.** See Supplementary Data 7 for library input sequences. Protein domains were ordered as GeneArt String DNA Fragments (Invitrogen) with attB sites and cloned into pDONR221 using the BP clonase II enzyme mix (Invitrogen). Each pENTR221 protein was sub-cloned into pAWH by Gateway recombination with LR clonase II enzyme (Invitrogen). Equimolar amounts of each pAWH protein clone were mixed to produce a pAWH RNAlib pool.

**Construction of RNA library vector for rec-Y3H.** See Supplementary Fig. 3 for detailed information of all vectors. A cassette containing a PolIII promoter—2xMS2BS—PolIII terminator, amplified from pIIIA-MS2-2 (kindly provided by Marvin Wickens), was used to replace the FspI/NotI fragment of pBKWH by Gibson assembly reaction. This generated pMS22H (plasmid with MS2 Binding sites, kanamycin resistance, and homology regions 1 and 2).

**RNA library construction by Gibson cloning.** See Supplementary Data 7 for library input sequences. RNA motif sequences with overhangs (5′-gaactagtggatccc-XXX-ccgggcagcttgcatgcctg-3′, where XXX represents an RNA) were ordered as single-stranded DNA oligonucleotides from Sigma and cloned into the XmaI site of pMS22H by Gibson assembly. Briefly, 1 μg of RNA motif oligonucleotide was annealed to primer Oligo_gib_rev (5′-CAGGCATGCAAGCTG-3′), and filled in with DNA Polymerase I, Large (Klenow) Fragment (NEB) following the manufacture's indications. Double-stranded DNA was purified with the Qiagen Mini-Elute PCR Purification Kit (Qiagen) and cloned into XmaI linearised pMS22H by Gibson assembly. All vectors were diluted to 5 nM, and equal amounts of each were mixed to give a pMS22H-RNA pool at a final concentration of 5 nM.

**Yeast strains.** Y2HGold was purchased from Clontech. YBZ-1 was kindly provided by Dr. Marvin Wickens, University of Wisconsin-Madison. See Supplementary Table 2 for genotype and more details.

**rec-Y2H screening on agar plates.** All yeast media was prepared using Minimal SD base (Clontech) and Amino Acid Dropout mixes (Clontech), as indicated by Clontech. Agar plates were prepared by adding 20 g/l of Agar (Conda). Competent Y2HGold (Clontech) was prepared following Clontech's recommendations. To maximise the number of transformants, competent yeast was transformed the same day it was prepared. To minimise bias, 12 small-scale transformations were set up for each type of selection media used (RS and RIS). Each pDEST-ORF pool (pAWH-ORF pool and pBWH-ORF pool) was diluted to 5 nM. The day before yeast transformation, each pDEST-ORF pool was linearised with I-SceI and I-CeuI homing enzymes (New England Biolabs) at 37 °C for 16 h, followed by 20 min at 65 °C. For each transformation, 20 fmol of linear pAWH-ORF pool (approximately 100 ng) and 20 fmol of linear pBWH-ORF pool were co-transformed into competent Y2HGold cells following Clontech's protocol. Each transformation was resuspended in 200 μl of 0.9% NaCl, and all transformation reactions were pooled together. A small aliquot (100 μl) was saved to make serial dilutions for colony counting on 10-cm selection media agar plates. For each selection media (RS and RIS), four square BioAssay dishes, 245 × 245 mm (Corning) were used. For each plate, 600 μl of transformed yeast cells from the transformation pool were spread and grown at 30 °C for 60 h. Yeast colonies were independently harvested from the RS plates and RIS plates by scraping them off with a cell spreader and 2 × 20 ml of

PBS. Each pool of cells was spun down at 700 g for 5 min, and resuspended in PBS. Cell concentration was determined from the OD at 660 nm.

**rec-Y2H screen in Seaprep agarose liquid–gel media.** Unless stated otherwise, screening steps are identical to the ones described in the previous section. On the same day of transformation, 250 ml liquid–gel media was prepared for each selection media (RS and RIS) by adding Seaprep Agarose (Cultek) to a final concentration of 0.5% (w/v). After autoclaving, liquid–gel media was allowed to cool down at room temperature with constant stirring. For each selection media, 2.4 ml of the co-transformed Y2HGold pool was added to 250 ml of liquid–gel media, mixed by stirring, transferred to a 5 l flask, and incubated on ice for 1 h. Subsequently, flasks were carefully transferred to a 30 °C incubator, taking care that the gel matrix was not perturbed. Growth and Y2H screening was done for 60 h. Then, yeast colonies suspended in the liquid–gel were gently stirred and harvested by centrifugation at 1600 g for 10 min. The yeast cell pellet was washed with PBS to remove remaining agarose, spun down at 700 g for 5 min, and resuspended in PBS. Cell concentration was determined from the OD at 660 nm.

**rec-Y3H screen in Seaprep agarose liquid–gel media.** The rec-Y3H screen follows exactly the same protocol as the rec-Y2H screen with only three modifications. (1) Instead of the pBWH-ORF pool, the pMS22H-RNA library is used. (2) The YBZ-1 yeast strain (kindly provided by Marvin Wickens) is used instead of the Y2HGold strain. (3) Different dropout RIS media are used (see Supplementary Table 2).

**Library preparation for paired-end sequencing.** See Supplementary Fig. 5 and Supplementary Methods. Although rec-Y2H is described here, this step is basically identical for both rec-Y2H and rec-Y3H screening. Minor differences for rec-Y3H can be seen at the end of this section. Library DNA for paired-end sequencing was prepared in parallel for each selection media (RS and RIS media). We strongly recommend performing all steps of the following rec-YnH library preparation protocol in 1 day because we found that the extracted yeast plasmid DNA can degrade rapidly. Recombined yeast plasmid DNA, referred to as pFAB (plasmid fusion of activation domain and binding domain), was extracted with Zymoprep™ Yeast Plasmid Miniprep II (Zymo Research). Yeast plasmid DNA was heated at 65 °C for 20 min, to minimise degradation. The pFAB-ORF pool was sheared to a size of 1500 bp by Covaris ultrasonication using the following conditions: duty cycle, 2%; intensity, 5; cycles/burst, 200; time, 25 s. Immediately after Covaris, sheared DNA was circularised. First, DNA ends were repaired with NEBNext® End Repair Module (New England Biolabs). Then, DNA was purified using the MinElute PCR Purification Kit (Qiagen). Finally, DNA was circularised by an intramolecular ligation reaction using the Quick Ligation™ Kit (New England Biolabs). To minimise PCR bias, circular DNA was split into 10 independent PCR reactions, with Q5 High-Fidelity 2X Master Mix (New England Biolabs), and primers Pr4seq_F_TS_R1 (5′-CCCTACACGACGCTCTTCCGATCTGCGCTGC AGGTCGACGGATC-3′) and Pr4seq_R_TS_R2 (5′-TTCAGACGTGTGCT CTTCCGATCTGCAGCTCGAGCTCGATGGATC-3′). Primers were custom synthesised by Integrated DNA Technologies. These primers specifically amplify from a pool of random fragments, those circular fragments that contain 3′ ends of both a bait-ORF and a prey-ORF, and at the same time add R1 and R2 Illumina adaptors. The R1/R2 PCR product was purified with Agencourt AMPure XP beads (Beckman Coulter) at a ratio of 35 μl beads for every 50 μl of PCR product. To introduce P5/P7 Illumina adaptors and a multiplexing index sequence, the R1/R2 PCR product was used as a template in a PCR reaction with a final volume of 150 μl split into six reactions with Q5 High-Fidelity 2X Master Mix (New England Biolabs), and NEBNext® Multiplex Oligos for Illumina®, Index Primers Set 1 (New England Biolabs). P5/P7 PCR was purified with Agencourt AMPure XP beads (Beckman Coulter) at a ratio of 30 μl of beads for every 50 μl of PCR product. To prepare rec-Y3H library DNA for paired-end sequencing, the same steps as for rec-Y2H were followed. Recombined yeast plasmid DNA is referred as pFAM protein–RNA pool (plasmid Fusion of Activation Domain and RNA stem-loop). To specifically amplify those fragments containing the 3′ end of a protein domain and an RNA site, and to add R1/R2 adaptors, Pr4seq_MS22_R1_rev (5′-CCCTAC ACGACGCTCTTCCGATCTGCAGGCATGCAAGCTGCC-3′) was used instead of primer Pr4seq_F_TS_R1, in combination with Pr4seq_R_TS_R2 (see Supplementary Fig. 7).

**Paired-end sequencing.** Amplicons containing R1/R2 adaptors and P5/P7 attachment sequences were quantified by qPCR and sequenced with 2 × 150 bp paired-end reads using a MiSeq Reagent Kit v2 (Illumina).

**Filtering and mapping of MiSeq sequencing data.** See Supplementary Fig. 7. For rec-Y2H, raw paired-end reads (read 1 and read 2) from RS and RIS conditions were filtered for both $R1_{Y2H}$ and $R2_{Y2H}$ sequences for read 1 and read 2, respectively (see below). Trimmed reads that are shorter than 15 nucleotides in length were discarded. The underlined nucleotides indicate differences between the $R1_{Y2H}$ and $R2_{Y2H}$ sequences. Bold characters represent the common attB2 sequence (ACCACTTTGTACAAGA AAGCTGGGT).

R1$_{Y2H}$: <u>CGCTGCAGGTCGACGGATCTTAGTTACTT</u>**ACCACTTTGTACA AGAAAGCT GGGT**

R2$_{Y2H}$: <u>GCAGCTCGAGCTCGATGGATCTTAGTTACTT</u>**ACCACTTTGTA CAAG AAAGCTGGGGT**

Then, the trimmed read sequences were aligned to the 3′ reverse complement coding sequence (for the last 100 nts) of the target library using the blastn programme with the blastn-short option and an E-value cut-off of 1e$^{-8}$. We picked the best aligned target protein and further filtered out the reads that were reverse-oriented aligned or aligned far from the 3′ end (>10 nts). When both read 1 and read 2 sequences were properly mapped to the input library sequences, we counted them as usable reads for interaction mapping. Since our library consists of recombined baits and preys, and we are reading from constant plasmid parts into the 3′ ends of fused ORFs, we can reliably distinguish baits (as read 1) and preys (as read 2). For rec-Y3H, similarly, raw paired-end reads (read 1 and read 2) from RS and RIS conditions were filtered for both the R1$_{Y3H}$ and R2$_{Y3H}$ sequences for read 1 and read 2, respectively (see below), and trimmed reads shorter than 15 nucleotides in length were discarded. The underlined nucleotides indicate any differences. Bold characters represent the attB2 sequence (ACCACTTTGTACAAGAAAGCTGGGT) for the protein part (read 2).

R1$_{Y3H}$: <u>GCAGGCATGCAAGCTGCC</u>

R2$_{Y3H}$: <u>GCAGCTCGAGCTCGATGGATCTTAGTTACTT</u>**ACCACTTTGT ACAAGAA AGCTGGGT**

For the RNA part (read 1), we only used exactly matching cases given the short length of some target RNAs. The protein part (read 2) was mapped as for rec-Y2H.

**Assigning the IS from PPI pair reads**. To calculate IS for each pair, we first generated frequency matrixes for RIS ($F_{RIS}(x, y)$) and RS conditions ($F_{RS}(x, y)$), and normalised the number of mapped PPI reads to the total PPI reads counts (Eq. (1)). Then, we reconstructed the null matrix by multiplying the row and column-wise marginal frequency in RS media (Eq. (2)). These steps correct for stochastic detection errors between $x$ and $y$ pairs when library complexity is high.

$$F(x,y) = N(x,y)/\sum_{x,y \in ALL} N(x,y), \qquad (1)$$

where $N(x, y)$ represents the number of PPI read counts mapped for bait, $x$ and prey, $y$ proteins in a specific conditions. $F_{RS}$ stands for –W (RS condition) and $F_{RIS}$ stands for nutrient dropout (RIS condition). Then, we generated a null matrix from the marginal distributions to reduce the random selection bias (such as highly or lowly selected pairs) because of library complexity and low sequencing depth[5].

$$F_{\Phi}(x,y) = F_{RS}(x) \times F_{RS}(y) = \sum_{y \in ALL} F_{RS}(x,y) \times \sum_{x \in ALL} F_{RS}(x,y). \qquad (2)$$

$F_{RS}(x)$ and $F_{RS}(y)$ indicate the marginal probability of bait, $x$ and prey, $y$ proteins (i.e., row-wise and column-wise sum of the detection frequency $F_{RS}(x, y)$, respectively).

Frequency values for RIS ($F_{RIS}(x, y)$) were divided into two classes based on the probability of Gaussian fitting[14]. With non-zero frequency values, we used the R package "mixtools" to fit a bimodal (noise and signal) mixture model (normalmaxEM()) to identify the probability of each component. If the frequencies had more probability of belonging to noise than to signal, we set them to 0; otherwise, we used the real frequency values. In this way we could reduce the noise of the selection matrix ($F_{RIS}(x, y)$), assuming that the majority of the bait–prey combinations were not interacting.

The noise-filtered signal ($F_{RIS}'(x, y)$) is further normalised by the null matrix ($F_{\Phi}(x, y)$) to generate IS. This corrects for library amplification and clone overrepresentation. To remove stochastic false positives and increase sensitivity, several experiments are averaged. This generates the final average-IS matrix. Finally, the basal auto-activation signals are removed by subtraction of the upper quartile IS.

$$IS_{FINAL}(x,y) = IS(x,y) - Q3(IS(x,\cdot)). \qquad (3)$$

Q3 represents the upper quartile function that gives $(3n/4)^{th}$ value from $n$ total values. IS($x$, •) represents all the values belonging to a bait, $x$. In other words, the top 25th percentile IS for each bait protein. Usually, Q3(IS($x$, •)) is 0 for non-auto-activator proteins. The same pipeline is applied to process the data from both rec-Y2H and rec-Y3H screens.

When we access the signal-to-noise ratio, we calculated the ratio as following equation:

$$SN_{ratio} = IS_{pos}/IS_{neg}, \qquad (4)$$

where of $IS_{pos}$ and $IS_{neg}$ indicate interaction score of expected positive and negative interaction pairs, respectively.

**A priori detection of auto-activators**. Auto-activators were detected by calculating the ratio of bait protein-mapped read counts between RIS and RS conditions for an AD-empty (pBWH X76 pool/pAWH empty) and a full (pBWH X76 pool/

pAWH X76 pool) screen. Read count ratio was calculated as the number of reads in RIS conditions divided by the number of reads in RS conditions. We used an arbitrary cut-off which is a 2.5 fold higher read count ratio compared with mean read count ratio of all the proteins.

**Statistical analysis**. Unless specified otherwise, we tested the statistical significance (with Fisher Z-transformation) of the correlation coefficient $r$, using the stats.pearsonr function of the "scipy" python package. It converts a Pearson's correlation coefficient to an appropriate $t$-value. The statistical significance (two-tailed $P$-value) was then calculated using the approximation of Student's $t$-distribution with degrees of freedom $n-2$. For the $t$-test, we used the two-tailed $P$-value from the stats. ttest_ind function of the same package. As for the Hypergeometric test, we used the stats.hypergeom.cdf function in "scipy" package. Area under the curve (AUC) of receiver operating characteristic curves was calculated using the metrics.auc function of the "sklearn" python package. F1-Score and MCC were calculated by python script with the following formulas:

$$F1 - Score = \frac{(2 \times TP)}{(2 \times TP + FP + FN)}$$
$$MCC = \frac{TP \times TN - FP \times FN}{\sqrt{(TP+FP) \times (TP+FN) \times (TN+FP) \times (TN+FN)}}.$$

TP, FP, TN, and FN indicate true positive, false positive, true negative, and false negative, respectively.

**Plotting software**. The circular plot in Fig. 4c was generated with "circlize" library in R software[36]. All network visualisations were generated with Cytoscape[37]. Heatmaps such as in Fig. 4a were generated by "pheatmap" library in R software. Bar charts, scatter plots, and box plots were generated using the default function of R software. RNA secondary structures in Fig. 6f were generated with the "nupack" software package[38]. RNA–protein complex structures were generated with pymol. To generate figures, we first plotted them with R software in PDF format, and then merged them with Illustrator v15.1.

**Yeast two-hybrid spot test**. Selected pairs from the rec-Y2H screen were re-tested by individual pair transformations. Selected baits (pBWH-ORF-X) and preys (pAWH-ORF-Y) from the X76 screen (see Supplementary Data 4) were individually linearised by I-SceI and I-CeuI (New England Biolabs) digestion for 16 h at 37 °C followed by 20 min at 65 °C. Competent Y2HGold cells were prepared with the Frozen-EZ Yeast Transformation II Kit™ (Zymo Research) and aliquots (10 μl) were prepared in 96-deep-well plates. Cells were transformed with 4 fmol of linearised bait and 4 fmol of linearised prey plasmid following the manufacturer's instructions. Cells were grown in –Trp dropout media (RS media) for 21 h at 900 rpm, and then spotted on –Trp agar OmniTray plates (Thermo Fisher Scientific). After 3 days, spots were replicated on the indicated RIS agar plates.

**Yeast three-hybrid spot test**. Selected RNA motifs (pMS22H-RNA-x) and protein domains (pAWH-Prey-Y) from RNAlib were individually linearised by I-SceI and I-CeuI (New England Biolabs) digestion for 16 h at 37 °C followed by 20 min at 65 °C. Indicated pairs (10 fmol linear RNA and 10 fmol linear prey vector) were co-transformed in freshly prepared YBZ-1 competent yeast and spotted on –Trp agar OmniTray plates. After 3 days, spots were replicated on the indicated RIS media.

**NanoBRET**. An attR1-ccdB-attR2 cassette (from pBWH) was generated by PCR and cloned into the NotI site of pHTN HaloTag® CMV-neo (Promega) by Gibson assembly to generate pHTNW. An attR1-ccdB-attR2 cassette (from pAWH) was generated by PCR and cloned into the NotI site of pNLF1-N [CMV/Hygro] (Promega) by gibson, to give pNLF1W. Indicated ORFs were cloned into either pHTNW or pNLF1W by Gateway LR reaction (Thermo Fisher Scientific) following the manufacturer's indications. HEK293T cells were plated in 24-well plates at a density of $1.4 \times 10^5$ cells per well. Cells were transfected with 500 ng of pHTNW-ORF, 5 ng of pNLF1W-ORF, 0.75 μl of Lipofectamine 3000 and 1 μl of P3000 Reagent (Thermo Fisher Scientific). After 20 h, each transformation was re-plated in four wells of 96-well plates at a density of $1 \times 10^4$ cells per well for duplicate control and experimental samples (technical replicates), and PPIs were analysed with NanoBRET™ Nano-Glo® Detection System kit (Promega) following manufacturer's instructions. Each transformation experiment was performed at least twice. The corrected NanoBRET ratio was calculated according to the manufacturer's instructions.

**Code availability**. Computer code is available at https://github.com/lionking0000/recYnH/.

## Data availability

The raw data of X76/X71, X163, and RNA–protein screening have been submitted to the ArrayExpress short read database (http://www.ebi.ac.uk/arrayexpress) and assigned the identifiers E-MTAB-6953.

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

## Acknowledgements

We would like to thank Marvin Wickens for providing the Y3H plasmids and yeast strain, and acknowledge the staff of the Biomolecular Screening & Protein Technologies and the Genomics core facility for their assistance, especially to Jochen Hecht for fruitful discussions on the screening designs. Furthermore, we would also like to thank Erich Wanker for the Gateway cassette-containing vector and Martin Loose for providing clones of the PAR genes. We also acknowledge the support from the Spanish Ministry of Economy and Competitiveness (MINECO) for Juan de la Cierva-Incorporación Programme (IJCI-2014-22070) to J.-S.Y., L.S. (BFU2015-63571-P), and S.M. (BFU2014-54278-P and BFU2015-62550-ERC). We further acknowledge support of the Spanish Ministry of Economy and Competitiveness, "Centro de Excelencia Severo Ochoa 2013-2017", SEV-2012-0208 and the CERCA Programme/Generalitat de Catalunya. This work was funded by the Spanish Ministry of Economy, Industry and Competitiveness (MEIC) reference MINECO PE 2013-2016 PN FEDER and the European Regional Development Fund (ERDF). All sequencing was done in the CRG Genomics Core Facility.

## Author contributions

J.S.Y., M.G.C., S.P.M. conceived the project. M.G.C., S.P.M., J.S.Y. designed the experiments. M.G.C., N.L., C.C., K.B., V.B.S. performed the experiments. J.S.Y., S.P.M., M.G.C. analysed the data. S.P.M., J.S.Y., M.G.C., L.S. wrote the paper. S.P.M. coordinated the project.

## Additional information

**Competing interests:** The authors declare no competing interests.

