## [Peer Review File · Nature Communications]

Reviewers' comments:

Reviewer #1 (Remarks to the Author):

Report on Yang et al. „ rec-YnH: An assay for the many-by-many detection of direct protein-protein and protein-RNA interactions”.

Yang et al describe a Y2H and Y3H approach that uses in vivo recombination to create a plasmid that carries bait and prey sequences for protein interaction studies through illumina sequencing. At least three high profile approaches that enable Y2H analyses through a massive parallel sequencing readout were reported before (quoted in the manuscript 4-6) and this method somehow is within the spectrum of these methods. While Yachie et al and Trigg et al use CRE recombination in yeast to sequence prey-bait fusions as barcodes or ORFs respectively, this paper uses classical recombination in yeast. The big plasmids are isolated, sheared and circularized again to amplify bait –prey fusion pieces that occurred in the circularization step. As with other methods only a fraction (here 40%) does contain both prey and bait sequences and therefore provides useful information. Data are processed against a background of non-PPI selected recombined plasmids and filtered for autoactive/ sticky prey and bait constructs. A 71X71 experiment is carried out and benchmarked against literature PPIs. A subset of the 316 interaction was subject to pairwise retesting, which also included negative pairs. However a very high fraction of the non-interacting pairs also tested positive in the pair wise retest (25 / 90). A second test case of selected PAR and microtubulin proteins spans a 176x176 space. Finally, in a Y3H experiment with the MS2 coat protein system, i.e. testing RNA-protein interactions, 17 RNA targets and 19 proteins were cross-mated leading to 11 interactions at the given F1 cut-off (combined specificity and sensitivity).

The method presented is probably comparable to previously published work. It is unclear why it is better, which I personally think is not important as it is different in several aspects. Though the data are well analyzed and presented in terms of error statistics, there is no PPI validation attempt. The claim is that it is low-tech and affordable and thus scalable. Exactly this point, scalability, has not been demonstrated as both presented screens are not large in test space, rather both experiments are within the scope of an “ok size” one by one experiment. A novel aspect is the implementation of a large scale Y3H experiment.

Some specific points for consideration.

*) Test spaces of both experiments is small. The experiments do not suggest scalability.

*) What is confusing is the calibration spot test that results a false positive rate of 2.9%.

Fortunately that can not be right, there is a conceptual problem in this test: in a test space of 71x71 this rate would give 146 false positives, which is close to the number of actually detected PPIs (and much large in the 176x176 space: 234 found and 847 theoretically false positive). Non-interacting pairs do not have equal probability of resulting false positive signals. The tested space is too small to determine a false positive rate. With a false positive rate of 2.9% it will be impossible to go proteome wide!

*) It remains unclear to me whether the authors suggest to do two experiments, one with autoactivating constructs to see which ones are autoactivating and a new real one or whether the just suggest to subtract “the row-wise upper quartile score” in the attempt to compensate for basal auto-activation activity. Autoactivating constructs become a bigger and bigger problem when scaling up, they suck up all reads, they have to be removed. ... Are the AD-empty and BD-empty screens required? I believe yes, as the recombination is performed in bulk. This may mean that in order to remove autoactivating constructs pool construction has to start from PCR, the very beginning.

*) From just looking at the score distributions in Figure 3a and 4d the cut offs used seem very arbitrary. If we are looking for the most reliable interactions, F-score which includes sensitivity may not always be the best. For examples, in looking at the spot test results the authors turn to ROC curves.

*) The spot test is the only validation approach even though it is using the same method. It is well appreciated that the authors present their data in fig 11. Why do not all constructs recombine. In

the pairwise approach 31/192 do not grow on –Trp. On the other hand the X71 approach reports sampling and pair complexity reach 95.3% and 98.9%, respectively. This is seemingly a contradiction.

*) The novel part, the Y3H approach, results RNA protein interactions which are unfortunately not validated.

Reviewer #2 (Remarks to the Author):

In this study, the authors introduce a new yeast two- and three-hybrid-based pipeline for the many-by-many detection of protein-protein and protein-RNA interactions and use it to identify potential interactions between a selected group of proteins including PAR proteins, motor adaptors, microtubule-interacting proteins and RNA-binding proteins, and between RNA-binding proteins and RNAs. The concepts underlying the described approach are not new, but the developed pipeline is more simple and cost-effective than the existing methods. The effectiveness of the new pipeline is nicely supported by data, and, therefore, the paper will be interesting for a broad readership. I do not think that any additional experiments are necessary, but there are a number of points where the writing needs to be significantly improved.

Comments.

1. It appears that the authors are trying to downplay to some extent the overlap with other methods of many-by-many yeast two-hybrid screening approaches. For the reader, it would actually be much nicer if these methods were properly explained in the Introduction, and the more extensive comparison between the re-YnH methodology with the already published methods was provided in the Discussion.

2. The authors seem to systematically overstate what they have achieved concerning the actual discovery of interactions, and this needs to be profoundly changed. From the numerous high throughput and other yeast two-hybrid studies, it is clear that the yeast two hybrid method alone cannot “reveal” or “uncover” a novel interaction – it reveals a PUTATIVE novel interaction, which can be confirmed by subsequent work. Showing that two proteins can bind in artificial system does not prove that they actually do so in cells. This is an important distinction, because databases are full of such putative interactions, many of which were never confirmed or failed to be confirmed. Obviously, disproving that two proteins bind to each other is even more difficult than showing that they do, but this is an additional reason why the results of yeast two-hybrid assays, when not supported by biochemical data, should be treated with extreme caution. I do not think that the authors need to confirm the new potential interactions that they found by other means – but they do need to make it very clear throughout the manuscript including the abstract that the identified interactions are putative and that no follow up work of any kind has been done to show that any of them are real.

The authors have certainly not revealed “the architecture of previously published coprecipitated complexes” and this statement must be removed from the abstract. Just to give an example, the authors failed to pick up the well-established direct interactions between EB1 and its major partners, KIF2C and CLASP1/2, although there is very strong biochemical and mutagenesis data from different laboratories that these direct interactions exist. This clearly illustrates the limitation of the screen that needs to be discussed and indicates that one cannot regard the acquired dataset as anything that reveals in any quantitative way the actual organization of the +TIP complexes. So the statement “of the 98 EB-coprecipitated proteins we tested within 249 the X163 set, only seven directly bind to EBs” should be changed to “of the 98 EB-coprecipitated proteins we tested within 249 the X163 set, we detected seven direct interactions and failed to detect <insert number> previously found direct interactions”. The whole text of the manuscript needs to be carefully screened for similar overstatements, for example, to avoid creating an impression that the authors showed that EB2 really interacts with RNA-binding proteins. An alternative would obviously be to

perform the follow-up and show that the identified interactions are real, but it seems to me to be out of the scope of the current paper.

Reviewer #3 (Remarks to the Author):

In this manuscript Yang et al. describe a new yeast two-or three-hybrid-based screening approach that allows detection of interactions within protein libraries or between protein libraries and RNA fragment pools. Their approach, termed rec-YnH, relies on homologous recombination to generate bait-prey fusion libraries with high efficiency. Various improvements of yeast-two-hybrid (Y2H) assays, published in recent years, have extended the throughput and sensitivity of the assay, for example by coupling the readout to next-generation sequencing and generating chimeric bait-prey libraries in a multiplexed fashion. The authors here integrate past developments to introduce a Y2H assay that is less-laborious and more cost-effective. They also extend the assay by describing a new analysis pipeline to process the sequencing data that filters experimental noise and improves correlation between experiments. They demonstrate the use of rec-YnH by performing a series of different screens to identify interactors of PAR proteins, to identify interactions between EB proteins and +TIP proteins, and they further adapt their approach to map interactions between multiple RNA-binding proteins and RNAs. The approach would be of interest to researchers that want to implement yeast-two hybrid screens and also provides a resource for PAR and EB protein interactomes.

Overall, the manuscript is clearly written and presents a thorough evaluation of the new pipeline. The figures, methods and protocols provided are appropriately detailed. The manuscript also provides a wealth of data by performing rec-YnH screens for a variety of protein or protein-RNA groups. However, while the authors highlight and discuss some potentially interesting findings, they don't provide any independent confirmation of any of their novel findings. Some of the newly-detected interactors of PAR proteins could be verified by co-IP assays, or the hypotheses discussed in Figure 6f, about the RNA determinants directing the specificity of different RNA-binding domains, could be validated by in-vitro binding assays. Such validation would be important to support the ability of rec-YnH to detect specific interactions.

The authors should also provide any scripts for software used for their analysis pipeline.

Minor point: Reference 12 needs correction.

Reviewer #4 (Remarks to the Author):

In this paper, Yang et al describe an improved Yeast 2- and 3-hybrid screening pipeline, and demonstrate its utility for mapping protein domain interactions, for distinguishing direct and indirect interactions within protein complexes previously identified by co-immunoprecipitation, and for mapping interactions between multiple RNA binding proteins and RNAs.

Proteomic is not my area of expertise, so I will make no attempt to evaluate the method(s) presented here. Instead, as requested, I will restrict myself to commenting briefly on the significance of some of the biological findings. Focussing on the area in which I have more detailed knowledge, I would say that the analysis of multi-domain Par protein/motor protein/adaptor interactions shows significant promise. Showing that the screening pipeline can resolve a large fraction of the known domain-specific interactions among the PAR proteins gives some confidence in the overall power of the method. The identification of new interactions of PAR proteins with motor proteins and their adaptors opens up some exciting new avenues for further exploration. Progress in understanding how PAR proteins are localized in different contexts and

how they interact with various cellular effectors to elaborate distinct polarized states is currently limited by our fragmentary knowledge of interaction partners. The findings presented here suggest that it should be possible to screen rapidly and effectively through a large numbers of cell type specific binding interactions that can then be examined further for functional significance.

Point-by-point reply to the reviewer's comments

Reviewer #1 (Remarks to the Author):

Yang et al describe a Y2H and Y3H approach that uses in vivo recombination to create a plasmid that carries bait and prey sequences for protein interaction studies through illumina sequencing. At least three high profile approaches that enable Y2H analyses through a massive parallel sequencing readout were reported before (quoted in the manuscript 4-6) and this method somehow is within the spectrum of these methods. While Yachie et al and Trigg et al use CRE recombination in yeast to sequence prey-bait fusions as barcodes or ORFs respectively, this paper uses classical recombination in yeast. The big plasmids are isolated, sheared and circularized again to amplify bait –prey fusion pieces that occurred in the circularization step. As with other methods only a fraction (here 40%) does contain both prey and bait sequences and therefore provides useful information.

Author's reply: We agree with this assessment but would like to highlight that our method produces a significantly higher amount of usable reads (40-50%) compared to published methods as CrY2H-seq¹ (2.4%). Thus, rec-YnH does allow using sequencing capacities more efficiently. Furthermore, rec-Y2H discriminates both AD and BD fused proteins in 40-50% of reads *versus* 0.13% in the case of CrY2H-seq (see our **Fig. 2a and 6c**). This has two advantages; it increases the level of confidence if a pair is found in both orientations (please also see validations of novel interactions shown below – 11/13 interactions found in both orientations were validated, new **Supplementary Fig. 16**) and it allows to avoid auto-activation or toxicity induced limitations.

Unfortunately, we could not obtain information about the percentage of usable reads for BFG-Y2H². However, we note that homologous recombination in yeast with 60nt overlaps, used by rec-YnH, is highly efficient (>95% successful recombination rate³). This is considerably higher than the Cre-recombinase recombination rate (16-27%²) which was reported for BFG-Y2H and CrY2H-seq¹.

Data are processed against a background of non-PPI selected recombined plasmids and filtered for autoactive/sticky prey and bait constructs. A 71X71 experiment is carried out and benchmarked against literature PPIs. A subset of the 316 interaction was subject to pairwise retesting, which also included negative pairs. However a very high fraction of the non-interacting pairs also tested positive in the pair wise retest (25 / 90).

Author's reply: We apologize that the manuscript was not written clearly enough and agree that this number is high. It stems from the fact that we used the average of different stringency conditions to find the 163 positive interactions in the rec-Y2H X71 library screens, but we compared to less-stringent spot-test conditions (-Trp/+Aba). Comparing to most stringent spot-test conditions (-Trp/+Aba/-His/-Ade, **Supplementary Fig.11b**) and combining the two conducted spot tests (in total, 207 hetero-interaction and 30 homo-interaction cases, **Table S4**), lowers the false-negative count to 12. Among them, eight interactions not detected by rec-Y2H are homodimers. Similar to the CrY2H-seq method¹, rec-Y2H is currently less sensitive for homodimers, which is caused by less efficient PCR amplification of fragments containing identical coding sequences (new **Supplementary Fig. 14**). We now reanalysed the performance of rec-Y2H without homodimers, as done for CrY2H-seq¹. In summary, the number of non-interacting pairs tested positive (false negatives) in the pair wise retest was (4/122) for hetero-interactions. We updated the manuscript text to explain this entire analysis better.

A second test case of selected PAR and microtubulin proteins spans a 176x176 space. Finally, in a Y3H experiment with the MS2 coat protein system, i.e. testing RNA-protein interactions, 17 RNA targets and 19 proteins were cross-mated leading to 11 interactions at the given F1 cut-off (combined specificity and sensitivity). The method presented is probably comparable to previously published work. It is unclear why it is better, which I personally think is not important as it is different in several aspects. Though the data are well analyzed and presented in terms of error statistics, there is no PPI validation attempt.

Author's reply: We agree and have now added validations of novel interactions with a different method. We have tested 35 novel positive interactions and 5 negative interactions using the NanoBRET⁴ system. This system detects interactions based on a bioluminescence resonance energy transfer (BRET) assay within the cytoplasm of mammalian cells. We chose this system as it rules out two major causes for false-positives rec-YnH could possibly create (recruitment of RNA polymerase by BD-fused proteins or DNA-binding of AD-fuses proteins). Furthermore screening in the cytoplasm instead the nucleus provides a different protein-background, making false-positives through "bridging" proteins less likely. By NanoBRET, we could validate 21 of 35 new positives while 5 out of 5 negatives pairs were confirmed negative (see new **Supplementary Fig.16a** below). We observed that the percentage of NanoBRET-validated positive interaction pairs is increasing with higher average-IS (new **Supplementary Fig.16b** below) and pairs detected in both orientations (in rec-Y2H, all proteins are fused to the activation and binding domain. With "both orientations", we mean that an interaction was detected between AD-Protein1/BD-Protein2 and vice versa.) These observations allow using these screen read-outs as confidence indicators for putative novel interactions.

Supplementary Fig. 16. NanoBRET protein-protein interaction validation results. (a) NanoBRET validation experiment results. Tested protein-protein pairs were sorted by their average IS, decreasing from left to right. Red and blue bars represent positive and negative interaction controls, respectively. Pink and grey bars represent rec-Y2H positive interactions. Pink bars=above NanoBRET cut-off, grey bars = below NanoBRET cut-off. Light blue bars are rec-Y2H negative interactions. The cut-off was defined by the mean plus 2.3 standard deviations of NanoBRET scores of all negative interactions. Asterisk (*) indicates interactions detected above the cut-off (average IS=1.6) in both orientations (Px-AD – Py-BD or vice versa). (b) The percentages of NanoBRET-tested interaction pairs sorted by categories. rec-Y2H scores were binned into 5 different groups. The negative control and rec-Y2H negative interactions contained no NanoBRET positive pairs.

The claim is that it is low-tech and affordable and thus scalable. Exactly this point, scalability, has not been demonstrated as both presented screens are not large in test space, rather both experiments are within the scope of an “ok size” one by one experiment.

Author’s reply: We note the reviewers’ concern and have differentiated our statements about scalability in the manuscript. We can only make a positive statement about scalability in the range of 10^2 - 10^4 interactions. However we would like to highlight that the comparably efficient use of sequencing power of rec-YnH (40-50% of reads contain PPI information, please see out first reply above) and the development of interaction screening in easily scalable liquid-gel cultures offer the potential for great scalability. We do agree, scalability needs to be further explored in the future.

A novel aspect is the implementation of a large scale Y3H experiment.

Some specific points for consideration.

*) Test spaces of both experiments is small. The experiments do not suggest scalability.

Author’s reply: Please see our response above.

*) What is confusing is the calibration spot test that results a false positive rate of 2.9%. Fortunately that can not be right, there is a conceptual problem in this test: in a test space of 71x71 this rate would give 146 false positives, which is close to the number of actually detected PPIs (and much large in the 176x176 space: 234 found and 847 theoretically false positive). Non-interacting pairs do not have equal probability of resulting false positive signals. The tested space is too small to determine a false positive rate. With a false positive rate of 2.9% it will be impossible to go proteome wide!

Author’s reply: We agree that this part was not written clearly enough. We have now amended the text. The 2.9% referred to the 1 false positive out of 34 negative interactions (FP/(FP + TN)) found in a small-scale spot test (56-pairs) – indeed one cannot derive a rate from this small test set. Previously we described two spot tests (192-pairs and 56-pairs spot tests) separately, which might have been confusing. Now we combined both spot tests, leading to one false positive out of 119 negatives (1/119). We also tested 85 rec-Y2H-detected positive interactions (half of 163 positive interactions defined by rec-Y2H) and only one was not detected in individual spot tests, resulting in a false discovery rate of 1.2% (FP/(FP + TP), **Table S4d**).

*) It remains unclear to me whether the authors suggest to do two experiments, one with autoactivating constructs to see which ones are autoactivating and a new real one or whether the just suggest to subtract “the row-wise upper quartile score” in the attempt to compensate for basal auto-activation activity. Autoactivating constructs become a bigger and bigger problem when scaling up, the suck up all reads, they have to be removed. ... Are the AD-empty and BD-empty screens required? I believe

yes, as the recombination is performed in bulk. This may mean that in order to remove autoactivating constructs pool construction has to start from PCR, the very beginning.

Author's reply: We agree that this part was not clear enough and improved the writing. Indeed, the AD and BD-empty pre-screens are useful to remove auto-activators and use sequencing capacity more efficiently. We found that 54.8% of reads in the X76 library accounts for 3 auto-activators detected with AD-empty pre-screen, and removing them can save sequencing reads as mentioned by the reviewer. We conducted a BD-empty screen to test for toxicity of proteins that can occur in the case of AD-fusions. We recommend the row-wise upper quartile subtraction to remove basal auto-activation signals regardless of the pre-screening as we found that basal auto-activators can be dominant when strong ones were removed as described previously². We show below that upper-quartile subtraction decreased the basal auto-activator bait signals (horizontal lines), indicating that quartile-subtraction could reduce false-positives. Re-assembling the screen library without auto-activators, however, does not require any PCR-steps as mentioned by the reviewer. We merely re-assemble the pooled library of pENTR ORF clones without the respective BD-fused auto-activating clones, and repeat a pooled Gateway LR reaction. This is also described in the manuscript now. Re-cloning of the BD-fused library without auto-activators requires 2-3 days.

a. X71 library quartile correction example

b. X163 library quartile correction example

*) From just looking at the score distributions in Figure 3a and 4d the cut offs used seem very arbitrary. If we are looking for the most reliable interactions, F-score which includes sensitivity may not always be the best. For examples, in looking at the spot test results the authors turn to ROC curves.

Author's reply: We appreciated this reviewer's point and amended the corresponding part of the text to make our use of the F1-score and ROC curves more clear. The intention was to show that the cut-off chosen based on the F1-score, calculated relative to the union of BioGrid and HiPPIE databases, also gives the best trade-off between sensitivity and specificity determined relative to pairwise re-tests which we show by ROC curves. As the reviewer mentioned, F1-Score gives more weights on the sensitivity than specificity. We tested both cut-off values from F1-Score and MCC and found that the F1-Score performs better when we compared with individual spot test results; both gave 99.2% specificity but using the F1-Score results in a higher sensitivity (95.5% vs 73.9%).

*) The spot test is the only validation approach even though it is using the same method. It is well appreciated that the authors present their data in fig 11. Why do not all constructs recombine. In the pairwise approach 31/192 do not grow on -Trp. On the other hand the X71 approach reports sampling and pair complexity reach 95.3% and 98.9%, respectively. This is seemingly a contradiction.

Author's reply: We agree, this seems contradictory. We used pooled LR reaction for cloning rec-Y2H. However, for the spot-test we individually clone each bait and prey by separate LR reactions. We have now sequence-validated the individual clones used for spot-test and found that a small fraction of the individual clones were not correct. We have now amended **Table S4** accordingly. Nine other

pairs correspond to proteins with a certain degree of toxicity. Lack of visible growth after 3 days on plate likely does not mean that the clone cannot survive at all. We believe we could detect these pairs with rec-Y2H, due to the higher sensitivity of sequencing based read-out, as described previously⁵. We note that **Supplementary Fig. 11** is a representative spot test experiment, but does not contain all spots tests performed. As mentioned in the figure legend, pairs that did not grow on SD/-Trp in a first attempt were re-transformed and tested again in separate spot test experiment. **Table S4** now includes results from all spot tests.

*) The novel part, the Y3H approach, results RNA protein interactions which are unfortunately not validated.

Author's reply: We agree and performed additional experiments to examine the interactions found by rec-Y3H further. We have mutated RNA-contacting residues of the dsRBM1 and 2 domains. Interestingly, a mutation of dsRBM1 affecting the glutamate which binds a cytosine in the target RNA-loop⁶ (E16A) caused a loss of binding specificity of this domain (see below and new **Supplementary Fig. 19**). This might indicate that E16 is responsible for directing dsRBM1 to specific A-form helices, while the rest of the domain mainly recognises double stranded RNAs. Mutating the residues of dsRBM2 that recognise the minor groove of the RNA-helix (S29A;H30A) cause a loss of its interaction with all 3 double-strand RNAs tested (#11, #12, and #13) (**Supplementary Fig. 19**). Furthermore, we validated by individual spot tests that wild type dsRBM2 can bind RNAs #11, #12, and #13 while dsRBM1 only binds to RNA #12 specifically.

We would like to highlight, that the validation set of RNA-protein complexes used contains 16 published co-structures⁷, one standard positive control (IRE-IRP) and 2 negative control proteins (MBP, GFP). We specifically detected 9 of these 17 positives, while only one protein (dsRBM2 of Adar2) binds more than its known interaction partner. dsRBM domains, however, are known to bind A-form RNA helices with different sequences with different affinities⁶. We hence explain the two "novel interactions" of dsRBM2 with the known lack of affinity sensitivity of Y3H screens that likely leads to the recognition of multiple A-form RNA helices by dsRBM2 in our screen.

Supplementary Fig. 19. Mutations affecting dsRBM2 and dsRBM1 RNA binding specificity and overall activity (a) Mutated amino acids. (b) 10 fmol of linear pMS22H-RNA and 10 fmol of linear pAWH-Prey were co-transformed into competent YBZ-1 cells and spotted on SD/-Trp and SD/-Trp/-His +1 mM 3-AT and grown for 4 days. See **Table S7** for input sequences. (c) Interaction matrix showing the obtained results with grey filled boxes representing interactions.

Reviewer #2 (Remarks to the Author):

In this study, the authors introduce a new yeast two- and three-hybrid-based pipeline for the many-by-many detection of protein-protein and protein-RNA interactions and use it to identify potential interactions between a selected group of proteins including PAR proteins, motor adaptors, microtubule-interacting proteins and RNA-binding proteins, and between RNA-binding proteins and RNAs. The concepts underlying the described approach are not new, but the developed pipeline is more simple and cost-effective than the existing methods. The effectiveness of the new pipeline is nicely supported by data, and, therefore, the paper will be interesting for a broad readership. I do not think that any additional experiments are necessary, but there are a number of points where the writing needs to be significantly improved.

Comments.

1. It appears that the authors are trying to downplay to some extent the overlap with other methods of many-by-many yeast two-hybrid screening approaches. For the reader, it would actually be much nicer if these methods were properly explained in the Introduction, and the more extensive comparison between the re-YnH methodology with the already published methods was provided in the Discussion.

Author's reply: We have amended the introduction and discussion to give a more transparent comparison of available methods and ours.

2. The authors seem to systematically overstate what they have achieved concerning the actual discovery of interactions, and this needs to be profoundly changed. From the numerous high throughput and other yeast two-hybrid studies, it is clear that the yeast two hybrid method alone cannot “reveal” or “uncover” a novel interaction – it reveals a PUTATIVE novel interaction, which can be confirmed by subsequent work. Showing that two proteins can bind in artificial system does not prove that they actually do so in cells. This is an important distinction, because databases are full of such putative interactions, many of which were never confirmed or failed to be confirmed. Obviously, disproving that two proteins bind to each other is even more difficult than showing that they do, but this is an additional reason why the results of yeast two-hybrid assays, when not supported by biochemical data, should be treated with extreme caution. I do not think that the authors need to confirm the new potential interactions that they found by other means – but they do need to make it very clear throughout the manuscript including the abstract that the identified interactions are putative and that no follow up work of any kind has been done to show that any of them are real.

Author's reply: We agree and have changed the manuscript accordingly. In addition we have used a different method to validate our results and to test the significance of novel putative interactions detected by rec-Y2H. We also agree and state so in the introduction, that Y2H-based screens can function as a generator for hypothesis that need to be tested in following experiments in cells or with pure components. Still we would like to emphasise, that our article contains three validation experiments (**Figures 3, 4, and 6**). In these experiments, we test the specificity and sensitivity of our method on interaction datasets that have been validated in the literature through several different experimental approaches. The X76 library was build based on extensively characterised interactions², PAR-protein-domain interactions we detected where all described in the literature and the RNA-protein interactions that we detect with high specificity were shown by co-complex structures and independent biophysical measurements⁷. We believe that the good performance our method shows on these three different test-sets allows to put a high degree of trust in the obtained results. The high validation rate we have now added with screens that are based on a different read-out in a different environment further strengthens the confidence of rec-YnH results (please see new **Supplemental Fig. 16** and the response to Reviewer 1 above).

The authors have certainly not revealed “the architecture of previously published coprecipitated complexes” and this statement must be removed from the abstract. Just to give an example, the authors failed to pick up the well-established direct interactions between EB1 and its major partners, KIF2C and CLASP1/2, although there is very strong biochemical and mutagenesis data from different laboratories that these direct interactions exist. This clearly illustrates the limitation of the screen that needs to be discussed and indicates that one cannot regard the acquired dataset as anything that reveals in any quantitative way the actual organization of the +TIP complexes. So the statement “of the 98 EB-coprecipitated proteins we tested within 249 the X163 set, only seven directly bind to EBs” should be changed to “of the 98 EB-coprecipitated proteins we tested within 249 the X163 set, we

detected seven direct interactions and failed to detect previously found direct interactions”. The whole text of the manuscript needs to be carefully screened for similar overstatements, for example, to avoid creating an impression that the authors showed that EB2 really interacts with RNA-binding proteins. An alternative would obviously be to perform the follow-up and show that the identified interactions are real, but it seems to me to be out of the scope of the current paper.

Author’s reply: We agree and changed the wording in the corresponding passages. It is now clearly stated that our screen is highly specific but on the datasets tested sensitivity is between 20-40%, which leads to false-negatives as the mentioned KIF2C and CLASP1/2 interactions of EB1. As discussed in the answer to Reviewer 1 above, yeast two hybrid screens can only reveal a fraction of all positive interactions. We still hope that our method can be used complementarily to mass spec analysis to predict the putative architecture of the pulled-down complex.

Reviewer #3 (Remarks to the Author):

In this manuscript Yang et al. describe a new yeast two- or three-hybrid-based screening approach that allows detection of interactions within protein libraries or between protein libraries and RNA fragment pools. Their approach, termed rec-YnH, relies on homologous recombination to generate bait-prey fusion libraries with high efficiency. Various improvements of yeast-two-hybrid (Y2H) assays, published in recent years, have extended the throughput and sensitivity of the assay, for example by coupling the readout to next-generation sequencing and generating chimeric bait-prey libraries in a multiplexed fashion. The authors here integrate past developments to introduce a Y2H assay that is less-laborious and more cost-effective. They also extend the assay by describing a new analysis pipeline to process the sequencing data that filters experimental noise and improves correlation between experiments. They demonstrate the use of rec-YnH by performing a series of different screens to identify interactors of PAR proteins, to identify interactions between EB proteins and +TIP proteins, and they further adapt their approach to map interactions between multiple RNA-binding proteins and RNAs. The approach would be of interest to researchers that want to implement yeast-two hybrid screens and also provides a resource for PAR and EB protein interactomes.

Overall, the manuscript is clearly written and presents a thorough evaluation of the new pipeline. The figures, methods and protocols provided are appropriately detailed. The manuscript also provides a wealth of data by performing rec-YnH screens for a variety of protein or protein-RNA groups. However, while the authors highlight and discuss some potentially interesting findings, they don’t provide any independent confirmation of any of their novel findings.

Author’s reply: We agree and have confirmed novel interactions using the NanoBRET⁴ system. This system detects interactions based on a bioluminescence resonance energy transfer (BRET) assay within mammalian cells (Please find details in the response to Reviewer 1 above and in the new **Supplementary Fig. 16**). We chose NanoBRET, as it rules out two major causes for false-positives rec-YnH could possibly create (recruitment of RNA polymerase by BD-fused proteins or DNA-binding of AD-fuses proteins). Furthermore screening in the cytoplasm instead the nucleus provides a different protein-background, making false-positives through “bridging” proteins less likely. By NanoBRET we could validate 21 of 35 new positives while 5 out of 5 negatives pairs were confirmed negative. We observed that the percentage of NanoBRET-validated interaction pairs is increasing with higher average-Interaction Score and pairs detected in both orientations (in rec-Y2H, all proteins are fused to the activation and binding domain. With “both orientations”, we mean that an interaction was

detected between AD-Protein1/BD-Protein2 and vice versa). These observations allow using the mentioned screen read-outs as confidence indicators for putative novel interactions.

Some of the newly-detected interactors of PAR proteins could be verified by co-IP assays, or the hypotheses discussed in Figure 6f, about the RNA determinants directing the specificity of different RNA-binding domains, could be validated by in-vitro binding assays. Such validation would be important to support the ability of rec-YnH to detect specific interactions.

Author's reply: We agree and tested several new putative interactions of PAR proteins by NanoBRET⁴. Please see newly added **Supplementary Fig. 19** and the response to Reviewer 1 above. To test the hypothesis discussed in **Fig. 6f**, we choose to mutate dsRBM1/2 residues predicted to be essential for RNA recognition⁶ (see new **Supplementary Fig. 19** and above). We find that altering both domain sequences either causes loss of binding specificity or loss of binding activity. When we mutated the RNA-loop contacting glutamate-16 of dsRBM1 to alanine, dsRBM1 loses its target specificity, now recognising all three target RNAs tested (#11, #12, and #13). In the case of dsRBM2, we mutated the RNA-helix minor groove contacting residues (S29A;H30A) and found that this causes the loss of interactions with all three double stranded RNAs tested (#11, #12, and #13).

The authors should also provide any scripts for software used for their analysis pipeline.

Author's reply: We have prepared a script and documentation on how to use it which will be available for public download on <https://github.com/lionking0000/recYnH>. The program was written in python and R. Required libraries are specified online.

Minor point: Reference 12 needs correction.

Author's reply: We apologise and have now corrected the reference.

Reviewer #4 (Remarks to the Author):

In this paper, Yang et al describe an improved Yeast 2- and 3-hybrid screening pipeline, and demonstrate its utility for mapping protein domain interactions, for distinguishing direct and indirect interactions within protein complexes previously identified by co-immunoprecipitation, and for mapping interactions between multiple RNA binding proteins and RNAs.

Proteomic is not my area of expertise, so I will make no attempt to evaluate the method(s) presented here. Instead, as requested, I will restrict myself to commenting briefly on the significance of some of the biological findings. Focussing on the area in which I have more detailed knowledge, I would say that the analysis of multi-domain Par protein/motor protein/adaptor interactions shows significant promise. Showing that the screening pipeline can resolve a large fraction of the known domain-specific interactions among the PAR proteins gives some confidence in the overall power of the method. The identification of new interactions of PAR proteins with motor proteins and their adaptors opens up some exciting new avenues for further exploration. Progress in understanding how PAR proteins are localized in different contexts and how they interact with various cellular effectors to elaborate distinct polarized states is currently limited by our fragmentary knowledge of interaction partners. The findings presented here suggest that it should be possible to screen rapidly and effectively through a large numbers of cell type specific binding interactions that can then be examined further for functional significance.

Author's reply: We thank the reviewer for the positive evaluation on the usefulness of our method.

References

1. Trigg SA, *et al.* CrY2H-seq: a massively multiplexed assay for deep-coverage interactome mapping. *Nature methods* **14**, 819-825 (2017).
2. Yachie N, *et al.* Pooled-matrix protein interaction screens using Barcode Fusion Genetics. *Molecular systems biology* **12**, 863 (2016).
3. Hua SB, Qiu M, Chan E, Zhu L, Luo Y. Minimum length of sequence homology required for in vivo cloning by homologous recombination in yeast. *Plasmid* **38**, 91-96 (1997).
4. Machleidt T, *et al.* NanoBRET-A Novel BRET Platform for the Analysis of Protein-Protein Interactions. *ACS Chemical Biology* **10**, 1797-1804 (2015).
5. Weimann M, *et al.* A Y2H-seq approach defines the human protein methyltransferase interactome. *Nature methods* **10**, 339-342 (2013).
6. Stefl R, *et al.* The solution structure of the ADAR2 dsRBM-RNA complex reveals a sequence-specific readout of the minor groove. *Cell* **143**, 225-237 (2010).
7. Yang X, Li H, Huang Y, Liu S. The dataset for protein-RNA binding affinity. *Protein science : a publication of the Protein Society* **22**, 1808-1811 (2013).

REVIEWERS' COMMENTS:

Reviewer #1 (Remarks to the Author):

Revised version: Yang et al. „ rec-YnH: An assay for the many-by-many detection of direct protein-protein and protein-RNA interactions”.

I think the manuscript improved substantially. It is very interesting both the methods aspects presented and the results obtained. The revised version indeed describes the auto-activator issue in clear detail in the data analysis now, which can be interpreted as a critical account on the method and makes the description more useful. For researchers to try it is much clearer what to expect now.

Reviewer #2 (Remarks to the Author):

The authors have adequately addressed my comments and I support publication of this manuscript

Reviewer #3 (Remarks to the Author):

The authors have improved the manuscript and have addressed all my concerns.